# NC-BENCH AND NCFOLD: A BENCHMARK AND CLOSED-LOOP FRAMEWORK FOR RNA NON-CANONICAL BASE-PAIR PREDICTION

**Heqin Zhu**[1,2*], **Ruifeng Li**[3,4*], **Ao Chang**[1,2], **Mingqian Li**[4],
**Hongyang Chen**[4†], **Peng Xiong**[1,2,5†], & **S. Kevin Zhou**[1,2,5,6,7†]

[1]School of Biomedical Engineering, Division of Life Sciences and Medicine, University of Science and Technology of China (USTC)
[2]MIRACLE Center, Suzhou Institute for Advance Research, USTC
[3]College of Computer Science and Technology, Zhejiang University
[4]Zhejiang Lab
[5]Biomedical Basic Research Center (BBRC) of Jiangsu Province
[6]State Key Laboratory of Precision and Intelligent Chemistry, USTC
[7]Jiangsu Provincial Key Laboratory of Multimodal Digital Twin Technology
`{zhuheqin1, ruifeng.li.99}@gmail.com`
`dr.h.chen@ieee.org, {xiongxp, skevinzhou}@ustc.edu.cn`

## ABSTRACT

RNA secondary structure forms the basis for folding and function, with non-canonical (NC) interactions indispensable for catalysis, regulation, and molecular recognition. Despite their importance, predicting NC base pairs remains challenging due to the absence of a standardized benchmark for systematic evaluation. To address this, we introduce **NC-Bench**, the first benchmark dedicated to NC base-pair prediction. NC-Bench provides 925 curated RNA sequences with 6,708 high-quality NC annotations, fine-grained edge and orientation classification tasks, and IsoScore-based embedding evaluation, offering a rigorous foundation for systematic assessment. Building on this, we propose **NCfold**, a dual-branch framework that couples sequence features with structural priors derived from RNA foundation models (RFMs) via Representative Embedding Fusion (REF) and REF-weighted self-attention. The closed-loop design iteratively refines sequence and structure representations, alleviating data sparsity and enhancing predictive accuracy. Experiments on NC-Bench show that NCfold outperforms existing methods, with zero-shot and ablation studies confirming its effectiveness and underscoring the need for NC-specific benchmarks. Together, NC-Bench and NCfold establish a systematic foundation for NC base-pair prediction, advancing our understanding of RNA structure and enabling next-generation RNA-centric applications. The datasets and codes are publicly available at https://github.com/heqin-zhu/NCBench.

## 1 INTRODUCTION

RNA molecules play essential roles in life, not only as carriers of genetic information but also as regulators, catalysts, and mediators of molecular interactions through their complex folding (Caprara & Nilsen, 2000; Cooper et al., 2009). The secondary structure of RNA forms the foundation for tertiary folding and interaction of other molecules, and further determines its function, which is largely dominated by base-pairing patterns (Seetin & Mathews, 2012; Zhang et al., 2025). Beyond canonical Watson-Crick (A-U and G-C) and wobble base pair (G-U), non-canonical (NC) base pairs (e.g., Hoogsteen edge) and their cis and trans orientations are widespread in natural RNAs (Mladek et al., 2009). In ribozymes, riboswitches, and long non-coding RNAs, NC base pairs frequently

---

*Equal contributions
†Corresponding authors

participate in tertiary interactions that are essential for their structural stability and biological function (Serganov & Patel, 2007). Thus, NC base pairs are not mere exceptions but rather critical contributors to RNA structural diversity and functional specificity (Olson et al., 2019).

Despite their importance, current computational methods (Zhao et al., 2021; Seetin & Mathews, 2012) remain limited in predicting NC base pairs, largely due to the lack of a standardized benchmark to support systematic development and evaluation. For instance, thermodynamic models (e.g., RNAstructure (Reuter & Mathews, 2010)), alignment-based methods (e.g., TurboFold II (Tan et al., 2017) and PFold (Seemann et al., 2008)), and deep learning-based methods (e.g., MXfold (Singh et al., 2019), UFold (Fu et al., 2022), and BPfold Zhu et al. (2025b)) are primarily designed for canonical base pairs or simple sequence-based non-canonical base pairs, with no or minimal coverage of geometrically details such as pair orientation and pair edge type. The absence of such a NC benchmark is itself non-trivial: annotating NC base pairs requires high-resolution 3D RNA structures (Olson et al., 2019), which are nevertheless extremely scarce, and the NC pairs are highly imbalanced in distribution across types (Stombaugh et al., 2009). Furthermore, biologically meaningful metrics that reflect the geometric and functional complexity of NC interactions are still lacking (Parisien et al., 2009). These challenges have long hindered NC pairs from being benchmarked and highlight the urgent need for a unified, fair framework to drive progress.

To bridge the gap, we present NC-Bench, the first standardized benchmark for NC base-pair prediction. We systematically curate a dataset of 925 high-quality RNA sequences from the Protein Data Bank (PDB) (Berman et al., 2000), annotated with 6,708 NC base pairs using tool RNAVIEW (Yang et al., 2003). Following the Leontis-Westhof scheme (Leontis & Westhof, 2001), we extract NC base pairs and exclude canonical Watson-Crick and wobble pairs, resulting in the largest systematic collection of its kind. NC-Bench defines fine-grained classification tasks for base-pair edges (Hoogsteen, Sugar, Watson-Crick) and orientations (cis, trans). It employs a standardized 4-fold cross-validation protocol and multiple metrics to ensure fair, biologically grounded evaluation, which enables fair, biologically grounded evaluation and provides a solid foundation for method development and comparison. In addition to standard accuracy-based metrics, NC-Bench incorporates the IsoScore metric (Rudman et al., 2021), which assesses the isotropy and information richness of RFM embeddings. Through its comprehensive design, NC-Bench establishes a solid foundation for advancing RNA structure modeling and further biological applications.

Furthermore, based on NC-Bench, we propose **NCfold**, a novel framework designed to tackle the challenges of predicting NC base pairs, particularly the issue of data scarcity. To elaborate this challenge, we leverage RFM embeddings as contextual priors, providing rich structural information beyond the small set of labeled NC pairs. Specifically, NCfold adopts a dual-branch architecture that integrates base-to-base interaction priors derived from RFMs. By ranking RFM embeddings with the IsoScore, we select the top-k most informative ones, which are then fused through Representative Embedding Fusion (REF) for downstream processing. These fused embeddings are converted into pairwise interaction matrices and incorporated into the model's attention mechanism via REF-weighted self-attention. This mechanism establishes a feedback loop in which sequence-level features and structural priors iteratively refine one another, enabling more accurate NC base-pair predictions. Moreover, NCfold employs a multi-task learning strategy to jointly predict both edge types and orientations of base pairs, capturing the intrinsic relationship of NC pairs, thereby improving performance and generalization across diverse RNA structures. To validate its effectiveness, we evaluate NCfold on NC-Bench, where it consistently outperforms traditional baselines and state-of-the-art RFMs in both edge and orientation prediction. In addition, zero-shot comparisons with canonical-focused methods show that NCfold achieves more accurate pairing-status predictions, highlighting the necessity and value of NC-Bench. Together, our contributions can be summarized in three parts:

- We construct **NC-Bench, the first standardized benchmark targeting RNA non-canonical base-pairs**, which comprises 925 RNA sequences with 6,708 curated annotations for fine-grained edge and orientation tasks.

- We propose **NCfold, an NC pair prediction framework** that combines sequential features with RFMs-derived structural priors through Representative Embedding Fusion (REF) and REF-weighted self-attention.

- We conduct **comprehensive experiments with evaluation in five different metrics on NC-Bench**, comparing 7 traditional machine learning methods and 7 RNA language mod-

els in different settings, followed by an ablation study of NCfold. Moreover, we conduct zero-shot assessments on four existing RNA canonical base pair prediction methods for reference.

## 2 NC-BENCH: NEW BENCHMARK FOR TRAINING AND EVALUATION

### 2.1 DEFINITION OF NON-CANONICAL BASE PAIR

RNA base-pairing interactions are highly diverse, extending far beyond the Watson-Crick pairs (A-U and G-C) and the wobble (G-U) pair that dominate helical regions. The geometric classification proposed by Leontis and Westhof (Leontis & Westhof, 2001) provides a comprehensive framework to describe this diversity. In this scheme, each base presents three distinct hydrogen-bonding edges, i.e., the *Watson-Crick* (W), *Hoogsteen* (H), and *Sugar* (S; also referred to as the shallow-groove) edges and non-edge category ($N_e$), while the relative orientation of the glycosidic bonds is classified as *cis*, *trans*, and non-pair ($N_p$). Combining the interacting edges of the two bases with their orientation yields 18 unique geometric categories of base pairs. Following the standard convention, *cis* WW families (including the G-U wobble variant) are treated as canonical, whereas all other edge-orientation combinations are regarded as *non-canonical* base pairs (NC pairs).

**Task Definition** In NC-Bench, given an RNA sequence $x = (x_1, \ldots, x_L)$, the objective is to predict the non-canonical base-pairing type between any nucleotide pair $(i, j)$, which is a nucleotide-level multiclass classification task. The task consists of two subtasks: edge prediction and orientation prediction. For edge prediction, we assign each nucleotide an edge label $e \in \{N_e, W, S, H\}$, resulting in an ordered edge pair $(e_i, e_j)$; after excluding canonical *cis* WW pairs (including the wobble G-U), this yields eight non-canonical edge combinations (WS, HS, SS, WH, SH, HH, HW, SW). For orientation prediction, each base pair is classified as either $o \in \{cis, trans\}$, with no base pairs labeled as $N_p$. Performances are evaluated mainly under per-class F1-score for both subtasks, along with additional assessments of generalization to RNA families.

### 2.2 BENCHMARK DATASET CONSTRUCTION

**Dataset Construction** We first collect 5,813 experimentally resolved RNA tertiary structures from the PDB database (Berman et al., 2000) with a cutoff date of 2023-08-09. We then apply RNAVIEW (Yang et al., 2003) to extract and identify base pairs from each PDB entry, filtering out other interactions such as stacking and 3D interactions, yielding 3,107 RNA structures with annotated base-pair interactions. To reduce sequence redundancy, we apply mmseqs (Steinegger & Söding, 2017) with parameters "–min-seq-id 0.8", "-c 0.8", and "–cov-mode 1", resulting in 1,673 non-redundant RNA sequences. Finally, we discard sequences that do not contain non-canonical base pairs, constructing the PDB-NC dataset of 925 RNA sequences containing NC pairs.

As shown in Figure 2a, we conduct a detailed analysis of the 6,708 non-canonical base-pair annotations identified across these 925 sequences. As defined by Leontis and Westhof, RNA base pairs span 1 canonical (WW) and 8 non-canonical (HH, SS, WH, HW, SH, HS, SW, and WS) edge-pair categories along with two kinds of orientations (trans and cis). Among them, WH pairs are the most frequent, accounting for approximately 26.94% of all NC pairs, followed by SH (18.50%), HW (13.80%), and SW (13.19%). In contrast, the proportions of WS, HS, and SS families are each below 9.00%, while HH pairs are the rarest, contributing only 3.40%. Regarding orientation, about 58.96% of NC pairs adopt the trans configuration, whereas 41.04% are cis. These statistics highlight the uneven distribution of NC pair families and orientations, reflecting the inherent biases and complexity of non-canonical interactions in natural RNA structures.

Finally, we employ 4-fold cross-validation on 925 samples, first splitting them in an 8:2 ratio into 740 training-validation samples and 185 fixed test samples. The training-validation set is further divided into four folds, with each fold serving once as validation while the others form the training set. After each round, the model is evaluated on both the validation fold and the fixed test set, yielding four test results whose average is reported as the final performance. The distributions of edge families and orientations are preserved, with detailed statistics in Appendix B.

**Evaluation Metrics** In NC-Bench, the prediction task is formulated as a multi-class classification problem with a limited dataset. To ensure a fair and comprehensive evaluation under such constraints, we employ five complementary metrics: Matthews correlation coefficient (MCC), accuracy (ACC), precision (P), recall (R), and F1-score (F1). Together, these metrics provide a comprehensive view of model performance in NC-Bench, covering overall accuracy, precision-recall balance, and robustness to class imbalance in limited data settings.

## 2.3 BENCHMARKING PROTOCOL FOR RFM EMBEDDINGS

To enable systematic and comparable evaluation of RNA representation learning models within the NC-Bench framework, we conduct a unified assessment of embeddings generated by a suite of state-of-the-art RNA foundation models (RFMs). Although these models demonstrate strong performance on their respective pretraining objectives, the embedding spaces they learn exhibit substantial heterogeneity in geometric structure and semantic distribution-some produce highly anisotropic representations, while others overly compress or semantic information. This distributional disparity implies that naively fusing embeddings from all models is unlikely to yield performance gains and may instead introduce noise or semantic inconsistency.

To address this issue, we adopt IsoScore (Rudman et al., 2021) as a quantitative metric for embedding quality. IsoScore effectively captures the degree of isotropy and uniformity of information density in the embedding space, thereby reflecting its generalization capacity and suitability for downstream tasks. Using this metric, we score and rank the embeddings from the following representative RFMs (RNA-FM (Chen et al., 2022), RNAErnie (Wang et al., 2024), SpliceBERT (Chen et al., 2024), UTR-LM (Chu et al., 2024), AIDO.RNA-650M (Zou et al., 2024), RiNALMo-micro (Penić et al., 2025), and structRFM (Zhu et al., 2025a)). We then select only the top-$k$ models according to their IsoScore rankings and fuse their embeddings via mean pool-

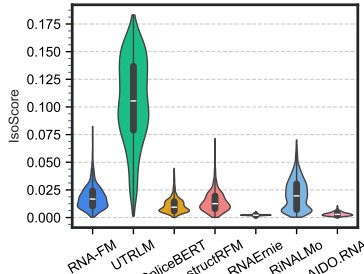

Figure 1: IsoScore distributions of RFMs.

ing to construct a more discriminative, robust, and semantically coherent joint representation, specifically tailored for non-coding RNA pair prediction.

## 3 METHOD

### 3.1 NCFOLD ARCHITECTURE

Non-canonical (NC) base pairs in natural RNAs are extremely sparse, making it infeasible to reliably learn their patterns from sequence-only self-attention. In other words, sequence information alone is insufficient. To address this, we propose NCfold (Figure 2c), a novel closed-loop dual-branch framework that integrates sequence context with structural priors derived from RFMs. Specifically, NCfold selects high-quality RFM embeddings using IsoScore, converts them into pairwise interaction matrices as structural interaction priors, and integrates them into the attention mechanism via REF-weighted self-attention. This yields a Dual-Branch Sequence-Matrix Framework, where priors guide sequence attention and, in turn, refining attention scores iteratively update the priors, enabling more robust NC base-pair prediction under data scarcity and sparse conditions.

**Notation** Let $x = (x_1, \ldots, x_L)$ denote an RNA sequence of length $L$, where each $x_i$ is a nucleotide token from the alphabet $\{A, U, C, G, N\}$, with "$N$" denoting an unknown nucleotide. Given a set of RNA foundation models (RFMs), we denote the token-level embedding from model $f$ as $\mathbf{E}^{(f)} \in \mathbb{R}^{L \times D_f}$, where $D_f$ denotes the hidden dimension of model $f$. To keep the efficiency of RFM embeddings, we select the top-$k$ embeddings by ranking the models based on IsoScore Rudman et al. (2021). The selected embeddings $\{\mathbf{E}^{(r)}\}_{r=1}^{k}$ are then converted into matrix features $\{\mathbf{M}^{(r)}\}_{r=1}^{k} \in \mathbb{R}^{L \times L}$ that encode nucleotide-to-nucleotide interactions by the "outer product mean" operation. These matrix features $\{\mathbf{M}^{(r)}\}_{r=1}^{k}$ are then stacked along the first dimension to yield a unified base-pair interaction representation $\mathbf{M} = \text{stack}(\{\mathbf{M}^{(r)}\}_{r=1}^{k}) \in \mathbb{R}^{k \times L \times L}$. Our prediction

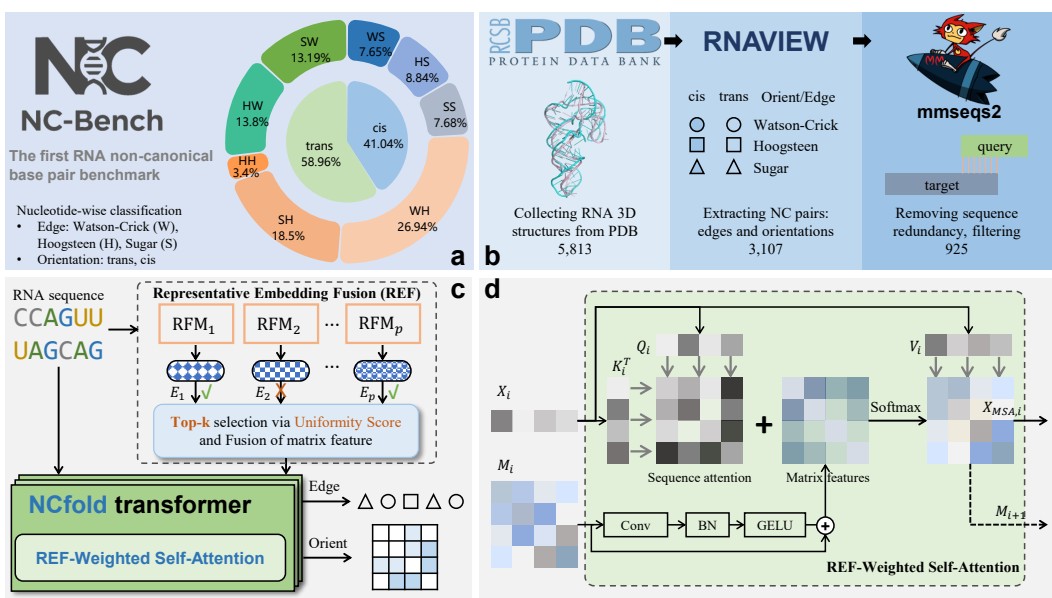

Figure 2: **Overview of NC-Bench and NCfold**. **a** The NC-Bench dataset is the first dataset for predicting RNA non-canonical base pairs, including three edges: Watson-Crick (W), Hoogsteen (H), and Sugar (S), forming eight kinds of pair-edge (HH, HW, SW, SH, WS, HS, SS, and WH) except canonical pair-edge "WW", and base-to-base orientations: trans and cis. **b** The pipeline of dataset curation. We firstly collect 5,813 RNA 3d structures from PDB database, then extracting NC pairs using RNAVIEW, successfully obtain 3,107 samples. Finally, we utilize mmseqs2 to remove sequence redundancy, and filter sequence with at least one NC-pair and a length no more than 512, obtaining 925 samples. **c** NCfold takes representative embedding fusion (REF) features and sequence as input, and predicts the edges and orientations of NC pairs. **d** The REF-weighted self-attention in NCfold transformer.

targets are given as $\hat{y}^{\text{edge}} \in \mathbb{R}^{L \times 4}$ with ground-truth $y^{\text{edge}} \in \{0, 1, 2, 3\}^L$, and $\hat{y}^{\text{orient}} \in \mathbb{R}^{L \times L \times 3}$ with ground-truth $y^{\text{orient}} \in \{0, 1, 2\}^{L \times L}$, where 0 represents no base-pair interaction.

**Representative Embedding Fusion (REF)**    To construct structural priors for NCfold, we introduce Representative Embedding Fusion (REF). The key idea is that not all embeddings from RFMs are equally informative: some are redundant or isotropic, while others capture richer structural patterns (Leclercq et al., 2009). Directly concatenating all embeddings would therefore introduce noise, especially under the sparse supervision of NC base pairs. Inspired by the Mixture of Experts (MoE) framework (Shazeer et al., 2017) and guided by the benchmarking results in Section 2, REF ranks RFM embeddings using IsoScore to quantify embedding quality and selects the top-$k$ most representative candidates. Each selected embedding $\mathbf{E}^{(r)} \in \mathbb{R}^{L \times D}$ is then converted into a pairwise interaction matrix by applying a mean operation to the outer product, yielding matrix feature $\mathbf{M}^{(r)} = \text{mean}(\mathbf{E}^{(r)} \otimes \mathbf{E}^{(r)\top}; -1, -2)$. The fused matrix representation is obtained via stacking as $\mathbf{M} = [\{\mathbf{M}^{(r)}\}_{r=1}^k] \in \mathbb{R}^{k \times L \times L}$. Unless otherwise stated, RFM neural network backbones remain frozen. This fused RFM embedding is then fed into the pairwise matrix branch of our dual-branch encoder and combined with the sequence branch for downstream edge and orientation prediction.

**REF-Weighted Self-Attention**    Non-canonical base pairs are extremely sparse in RNA structures, making it difficult for purely data-driven models to capture the diversity and complexity of base pairing interactions. To overcome this, we introduce REF-weighted self-attention, which injects structural priors from REF matrices directly into the attention mechanism.

The key idea is to bias the attention map toward biophysically plausible interactions while still allowing sequence features to adaptively refine them. Specifically, REF matrices $\mathbf{M}$ are processed through convolutional layers CONV to highlight local structural signals. The processed features

are then added to the raw attention scores $\mathbf{QK}^\top/\sqrt{d}$, ensuring that attention not only depends on sequence context but is also guided by REF priors. Formally, given sequence queries $\mathbf{Q}$, keys $\mathbf{K}$, and values $\mathbf{V}$, the REF-weighted attention is:

$$\text{REF-weighted Self-Attention}(\mathbf{X}) = \text{softmax}\left(\frac{\mathbf{QK}^\top + \text{CONV}(\mathbf{M})}{\sqrt{d}}\right)\mathbf{V}. \tag{1}$$

This mechanism creates a feedback loop: REF priors guide attention maps, attention scores refine the matrix features across layers. As a result, sequence and structure information are integrated in a closed-loop manner, improving robustness under data sparsity.

**Dual-Branch Sequence-Matrix Framework**  To better integrate sequence features with structural priors, we design a dual-branch sequence-matrix framework that enables iterative interactions across layers. In this design, sequence and matrix representations interact bidirectionally through cascaded REF-weighted self-attention blocks. Specifically, each transformer block in the cascade processes two types of inputs: the sequence feature $\mathbf{X}_i \in \mathbb{R}^{L \times d}$ and the attention weight matrix $\mathbf{M}_i \in \mathbb{R}^{h \times L \times L}$ from the previous block, where $h$ is the number of multi-head attention that enables parallel self-attention computations across different subspaces to enhance expressive power and efficient global context modeling.

In the $i$-th transformer block, the sequence branch first undergoes layer normalization and linear projections to generate queries, keys, and values:

$$\tilde{\mathbf{X}}_i = \text{LN}(\mathbf{X}_i), \quad \mathbf{Q}_i = \tilde{\mathbf{X}}_i \mathbf{W}_Q^{(i)}, \quad \mathbf{K}_i = \tilde{\mathbf{X}}_i \mathbf{W}_K^{(i)}, \quad \mathbf{V}_i = \tilde{\mathbf{X}}_i \mathbf{W}_V^{(i)}. \tag{2}$$

The matrix branch then incorporates the prior attention weights $\mathbf{M}_i$ (stacked REF $\mathbf{F}$ for the first block) into the current attention computation, which is firstly processed through a residual convolutional layer CONV. The REF-weighted attention is calculated as:

$$\mathbf{M}_{i+1} = \text{softmax}\left(\frac{\mathbf{Q}_i \mathbf{K}_i^\top + \text{CONV}(\mathbf{M}_i)}{\sqrt{d}}\right), \quad \mathbf{X}_{\text{MSA},i} = \mathbf{M}_{i+1}\mathbf{V}_i. \tag{3}$$

Following the multi-head attention operation, the sequence feature is updated through residual connection, layer normalization (LN), and feed-forward network (FFN):

$$\tilde{\mathbf{X}}_{\text{MSA},i} = \text{LN}(\mathbf{X}_{\text{MSA},i} + \tilde{\mathbf{X}}_i), \quad \mathbf{X}_{i+1} = \tilde{\mathbf{X}}_{\text{MSA},i} + \text{FFN}(\tilde{\mathbf{X}}_{\text{MSA},i}). \tag{4}$$

After $N$ cascaded transformer blocks with dual-branch dataflow, the output sequence feature $\mathbf{X}_N$ and the last attention weight matrix $\mathbf{M}_N$ capture refined base pairing patterns, ensuring that sequence-level contextual features and base-to-base interaction mutually reinforce each other across layers, further boosting the model building both local and global structural dependencies.

## 3.2 Loss Function

**Prediction Heads**  To produce task-specific outputs, the model employs two parallel projection layers. For the sequence pathway, a fully connected layer projects the final sequence embeddings to edge logits $\hat{\mathbf{Y}}_{\text{edge}} \in \mathbb{R}^{B \times L \times 4}$, obtaining output edge $\hat{\mathbf{Y}}_{\text{edge}} = \text{Linear}(\mathbf{X}_N)$, which is transposed as $\hat{\mathbf{Y}}_{\text{edge}} \rightarrow \hat{\mathbf{Y}}_{\text{edge}}^\top \in \mathbb{R}^{B \times 4 \times L}$ when the loss is computed to match the input format of cross-entropy. For the matrix pathway, a $1 \times 1$ convolution maps the fused interaction matrix to orientation logits $\hat{\mathbf{Y}}_{\text{orient}} \in \mathbb{R}^{B \times L \times L \times 3}$, obtaining the output orientation map $\hat{\mathbf{Y}}_{\text{orient}} = \text{Conv}_{1 \times 1}(\mathbf{M}_N)$, which is permuted to $\hat{\mathbf{Y}}_{\text{orient}}^\top \in \mathbb{R}^{B \times 3 \times L \times L}$ when the loss is computed.

**Loss Function**  Given ground-truth edge labels $\mathbf{Y}_{\text{edge}} \in \{0, 1, 2, 3\}^{B \times L}$ and orientation labels $\mathbf{Y}_{\text{orient}} \in \{0, 1, 2\}^{B \times L \times L}$, the training objective is defined as

$$\mathcal{L} = \lambda_{\text{edge}} \cdot \text{CE}\left(\hat{\mathbf{Y}}_{\text{edge}}^\top, \mathbf{Y}_{\text{edge}}\right) + \lambda_{\text{orient}} \cdot \text{CE}\left(\hat{\mathbf{Y}}_{\text{orient}}^\top, \mathbf{Y}_{\text{orient}}\right), \tag{5}$$

where CE denotes cross-entropy applied only to valid (non-padding) entries. In practice, we apply class-specific weights in both tasks ($[1, 5, 20, 20]$ for edges $[\text{N}_e, \text{W}, \text{H}, \text{S}]$ and $[1, 20, 20]$ for orientations $[\text{N}_p, \text{trans}, \text{cis}]$), which effectively alleviates the imbalance between frequent and rare categories. We further empirically set $\lambda_{\text{edge}} = 1$ and $\lambda_{\text{orient}} = 1$.

## 4 EXPERIMENT

During our experiments, we observed that both the number of layers in the model and the choice of batch size play a critical role in determining overall performance. As shown in Table 4, NCfold yields the most favorable outcomes when the number of transformer layers is set to 4 or 6. As shown in Table 5, the model achieves its best results when trained with a batch size of 4, suggesting that smaller batch sizes may help the model capture fine-grained patterns more effectively.

### 4.1 EXPERIMENTAL PROTOCOL

**Setup**   We train our NCfold according to the framework introduced in Section 3. For the sequence embedding, we consider $P = 7$ pre-trained RFMs, namely RNA-FM (Chen et al., 2022), RNAErnie (Wang et al., 2024), SpliceBERT (Chen et al., 2024), UTR-LM (Chu et al., 2024), AIDO.RNA-650m (Zou et al., 2024), RiNALMo-micro (Penić et al., 2025), and structRFM (Zhu et al., 2025a). The sources are listed in Appendix C. We adopt a 4-fold cross-validation strategy for training. The loss function is CrossEntropyLoss (Mao et al., 2023) with class weighting and label smoothing ($\varepsilon = 0.05$) to improve generalization. The model is optimized using AdamW (Loshchilov & Hutter, 2019) with a learning rate of 0.0001 and weight decay for regularization. Training runs for up to 60 epochs, with early stopping (patience=10) based on validation performance. Our implementation is based on PyTorch, running on a Linux server with an NVIDIA GeForce RTX 4090 (24 GB).

**Baseline Methods**   For the NC pair prediction task, we categorize the employed methods into two major groups, reflecting different perspectives on modeling RNA-related interactions: **1) Traditional machine learning approaches**, including Random Forest (Rigatti, 2017), Gradient Boosting (Natekin & Knoll, 2013), XGBoost (Chen et al., 2015), Stochastic Gradient Descent (SGD) (Bottou, 2012), Logistic Regression (LaValley, 2008), k-Nearest Neighbors (KNN) (Peterson, 2009), and Multilayer Perceptron (MLP) (Taud & Mas, 2017), which serve as strong baselines. **2) RNA foundation models (RFMs)**, the recently developed large-scale pretrained models, including RNA-FM (Chen et al., 2022) (pretrained on 24M sequences), RNAErnie (Wang et al., 2024), SpliceBERT (Chen et al., 2024), UTR-LM (Chu et al., 2024), AIDO.RNA-650M (Zou et al., 2024), RiNALMo-micro (Penić et al., 2025), and structRFM (Zhu et al., 2025a), which capture rich contextual and structural representations of RNA sequences. Their ability to leverage large corpora and learn transferable features makes them promising beyond conventional ML methods. To ensure fairness, we adopt identical hyperparameter settings across all experiments and employ 4-fold cross-validation for evaluation on the NC-Bench dataset. Further details on the baselines are provided in Appendix C.

### 4.2 RESULTS ANALYSIS ON NC-BENCH

Table 1 compares the performance of various baseline models on two NC pair prediction tasks: Pair-Edge and Orientation, using 4-fold cross-validation. In the Pair-Edge Prediction task, most traditional machine learning methods (such as Random Forest, Logistic Regression) perform poorly, with their MCC close to 0 and F1 ranging from 0.093 to 0.258, indicating limited performance. It should be noted that all RFMs achieve same metrics in the edge subtask, obtaining extremely low MCC, precision, recall and F1 (MCC = 0.000, P = 0.185, R = 0.281, F1 = 0.218), which is due to the failure prediction of positives. Specifically, the frozen RFM with a simple linear layer fails to predict the positive edges, resulting in all negatives ("no-edge"). In contrast, our NCfold series (especially NCfold top-2) outperforms all other baselines, showing a superior performance in the Pair-Edge prediction task. In the Orientation prediction task, all models show high ACC ($> 0.9$), which is due to the majority amount of negatives ("no-pair") and the larger total number ($L \times L$ for orientations, compared to $L$ for pair-edge). When comparing under precision, recall, and F1 metrics, our NCfold series still stand out in this task, especially NCfold (top-2), which significantly outperforms other models in key metrics like Precision (0.487), Recall (0.544), and F1 (0.486). Overall, NCfold (top-2) achieves the best performance in both tasks, showing clear and significant advantage in NC pair prediction. The statistics of REF frequency of RFMs are visualized in Appendix D.1.

We further utilize VARNA (Darty et al., 2009) to visualize three examples (PDB ID: 1G70, 1TLR, 2NC1) of reference structures and predicted structures by NCfold. The detailed NC edges (Watson-Crick, Hoogsteen, and Sugar) and their corresponding orientations (trans and cis) are displayed in Figure 3 and Appendix D.6. When comparing the prediction to reference structure, NCfold

Table 1: The mean metrics of baseline methods in two NC pair prediction tasks with 4-fold cross-validation. We use three colors of blue to denote the first, second, third best performance among naive machine learning methods and RFMs. It should be noted that F1 and MCC are more holistic, comprehensive metrics than others, especially in situation of imbalanced classification.

| Model | Pair-Edge | | | | | Orientation | | | | |
|-------|-----|-----|---|---|----|-----|-----|---|---|----|
| | MCC | ACC | P | R | F1 | MCC | ACC | P | R | F1 |
| Random Forest | -0.020 | 0.285 | 0.232 | 0.269 | 0.168 | 0.002 | 0.976 | 0.369 | 0.377 | 0.372 |
| Gradient Boosting | 0.042 | 0.554 | 0.274 | 0.293 | 0.258 | 0.002 | 0.924 | 0.371 | 0.379 | 0.365 |
| XGBoost | -0.008 | 0.439 | 0.246 | 0.265 | 0.221 | 0.000 | 0.979 | 0.376 | 0.385 | 0.380 |
| SGD | -0.028 | 0.178 | 0.106 | 0.211 | 0.101 | 0.111 | 0.792 | 0.396 | 0.539 | 0.369 |
| Logistic Regression | 0.007 | 0.106 | 0.096 | 0.173 | 0.093 | 0.026 | 0.050 | 0.327 | 0.434 | 0.037 |
| KNN | 0.014 | 0.345 | 0.255 | 0.279 | 0.188 | 0.000 | 0.979 | 0.376 | 0.385 | 0.380 |
| MLP | 0.000 | 0.638 | 0.185 | 0.281 | 0.218 | 0.163 | 0.939 | 0.395 | 0.472 | 0.402 |
| RNA-FM | 0.000 | 0.638 | 0.185 | 0.281 | 0.218 | 0.000 | 0.984 | 0.352 | 0.359 | 0.355 |
| structRFM | 0.000 | 0.638 | 0.185 | 0.281 | 0.218 | 0.005 | 0.972 | 0.358 | 0.361 | 0.357 |
| RNAErnie | 0.000 | 0.638 | 0.185 | 0.281 | 0.218 | 0.024 | 0.948 | 0.357 | 0.372 | 0.354 |
| SpliceBERT | 0.000 | 0.638 | 0.185 | 0.281 | 0.218 | 0.003 | 0.972 | 0.350 | 0.358 | 0.352 |
| UTR-LM | 0.000 | 0.638 | 0.185 | 0.281 | 0.218 | -0.001 | 0.983 | 0.352 | 0.358 | 0.355 |
| AIDO.RNA-650M | 0.000 | 0.638 | 0.185 | 0.281 | 0.218 | 0.003 | 0.963 | 0.358 | 0.359 | 0.355 |
| RiNALMo | 0.000 | 0.638 | 0.185 | 0.281 | 0.218 | 0.007 | 0.955 | 0.348 | 0.357 | 0.345 |
| NCfold (top-1) | 0.211 | 0.596 | 0.375 | 0.386 | 0.341 | 0.285 | 0.966 | 0.474 | 0.524 | 0.482 |
| NCfold (top-2) | 0.245 | 0.628 | 0.400 | 0.409 | 0.365 | 0.312 | 0.951 | 0.487 | 0.544 | 0.486 |
| NCfold (top-3) | 0.219 | 0.603 | 0.370 | 0.376 | 0.336 | 0.265 | 0.948 | 0.463 | 0.520 | 0.466 |
| NCfold (top-4) | 0.194 | 0.605 | 0.347 | 0.366 | 0.321 | 0.256 | 0.935 | 0.458 | 0.516 | 0.456 |

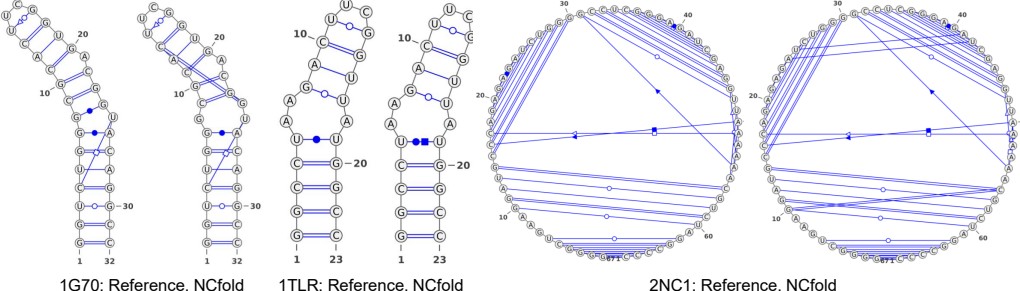

1G70: Reference, NCfold    1TLR: Reference, NCfold    2NC1: Reference, NCfold

Figure 3: Visualization of NC edges (Watson-Crick ○/●, Hoogsteen □/■, and Sugar △/▲) and corresponding orientations (trans/cis) from the reference structures and predictions by NCfold.

predicts accurate canonical base pairs, pseudoknot (intersecting lines), and NC base pair edges and orientations. The performance can be further improved once the model is trained on more canonical and pseudoknot annotations. Moreover, NCfold tends to predict more false positive pseudoknots, which can be relieved through postprocessing and adjusting the thresholds of the orientation map.

### 4.3 ZERO-SHOT EXPERIMENTS

Current methods for predicting the RNA secondary structure are inherently constrained to predicting only canonical base pairing status, namely "paired" and "non-pair" states, with NC pairings inferred solely from nucleotide information. Consequently, these existing methods cannot predict edge categories or orientations of base pairs. To evaluate and illustrate the gap in predicting NC pairings, we compare NCfold with four representative state-of-the-art RNA secondary structure prediction methods (MXFold2 (Sato et al., 2021), SPOT-RNA (Singh et al., 2019), UFold Fu et al. (2022), and BPfold (Zhu et al., 2025b)) using zero-shot assessments. These assessments focus explicitly on how

well each method predicts the pairing status of NC base pairs, which is framed as a binary classification task involving two mutually exclusive categories: "pair" and "non-pair". To extract NC pair predictions from the compared methods, we exclude all canonical base pairings, namely "A-U", "G-C", and "G-U" pairs. Notably, NCfold predicts the specific edge categories and orientations of NC pairs, at a level of detail that is not directly comparable to the outputs of pairing status by other methods. To enable a reference-based comparison, we convert NCfold 's predicted orientations (i.e., "trans" and "cis") to the "pair" category, aligning its output with the prediction format of existing methods.

As shown in Figure 4 (Table 6), existing methods (including BPfold, UFold, MXfold2, and SPOT-RNA) are designed to predict canonical base pairing status and thus fail to effectively predict NC pairing status. This failure results in only small numbers of positive predictions, leading to high precision but extremely low recall (approximately zero). In contrast, NCfold achieves the highest F1 score of 0.440 among all state-of-the-art methods evaluated, as it makes both accurate positive and negative predictions, yielding a precision of 0.489 and a recall of 0.431. Additionally, since the negative category (i.e., "non-pair") constitutes the majority of the $L \times L$ pairing space, all methods achieve high accuracy values greater than 0.950. Given the

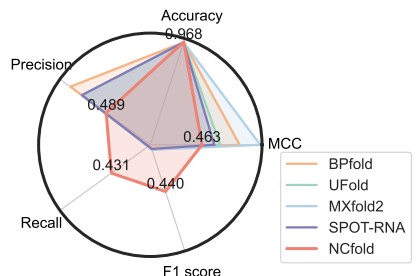

Figure 4: Zero-shot performances on pairing status ("pair" and "non-pair").

lack of existing benchmarks for NC pairs and methods tailored to NC pair prediction, NC-Bench fills this gap by serving as the first dataset for advancing RNA structure modeling and deepening biological understanding of RNA.

## 4.4 ABLATION STUDY

Table 2: Results of the ablation study. The best method is highlighted in bold. (NCfold-base: without incorporating RFMs embeddings; NCfold-BPE: with BPfold's base pair motif energy).

| Model | Pair-Edge | | | | | Orientation | | | | |
|---|---|---|---|---|---|---|---|---|---|---|
| | MCC | ACC | P | R | F1 | MCC | ACC | P | R | F1 |
| NCfold-base | 0.084 | 0.605 | 0.221 | 0.352 | 0.251 | 0.217 | 0.948 | 0.402 | 0.522 | 0.419 |
| NCfold-BPE | 0.211 | **0.656** | 0.358 | 0.402 | 0.335 | **0.326** | **0.966** | 0.435 | **0.562** | 0.464 |
| NCfold | **0.245** | 0.628 | **0.400** | **0.409** | **0.365** | 0.312 | 0.951 | **0.487** | 0.544 | **0.486** |

Table 2 shows the results of three models: NCfold-base, NCfold-BPE, and NCfold. NCfold-base represents the model without incorporating the RFMs structural prior, while NCfold-BPE incorporates BPfold's base-to-base energy priors. NCfold-base shows the worst performance across all metrics, with a low MCC (0.084) and F1 score (0.251) in the Pair-Edge prediction task, and the lowest MCC (0.217) in the Orientation prediction task. NCfold-BPE performs better than NCfold-base in both tasks, with a notable improvement in MCC (0.326) and F1 (0.464) for orientation prediction. However, it still lags behind NCfold, which incorporates RFMs for embedding and achieves the best overall performance. NCfold shows the highest F1 score in both tasks, with an F1 of 0.365 in Pair-Edge, and an F1 of 0.486 in Orientation. This clearly demonstrates that incorporating RFMs into the model leads to the most significant performance improvements.

## 5 CONCLUSION AND DISCUSSION

We present NC-Bench and NCfold, the first standardized benchmark and framework for non-canonical RNA base-pair prediction. Extensive experiments show that NCfold significantly outperforms both traditional machine learning baselines and state-of-the-art RFMs, while ablation and zero-shot studies highlight the importance of base-to-base interaction priors and confirm the necessity of NC-Bench for advancing RNA structure modeling.

Looking forward, we envision several directions for future work. First, while NC-Bench represents the largest curated dataset to date, its scale remains modest compared to canonical base-pair resources; extending the dataset with new experimental structures will further enhance benchmark robustness. Second, NCfold currently operates under a supervised learning paradigm; integrating semi-supervised or generative approaches could better exploit unlabeled RNA data. Finally, the closed-loop dual-branch design introduced here may be extended to other RNA-centric tasks, such as tertiary structure modeling, RNA-protein interaction prediction, or RNA design, thereby broadening the impact of this framework.

Conclusively, NC-Bench and NCfold establish a systematic foundation for advancing NC pair prediction, contributing both a rigorous evaluation standard and a novel methodological framework that deepens our understanding of RNA structural complexity and enables the development of next-generation RNA-centric applications. More discussion is provided in Appendix E.

## ACKNOWLEDGMENTS

This work is supported by National Natural Science Foundation of China (62501537 to H.C., 62271465 to S.K.Z., 32370581 to P.X.), Suzhou Basic Research Program (SYG202338 to S.K.Z.), and National Key Research and Development Program of China (2023YFB4502400 to H.C.).

## ETHICS STATEMENT

This paper proposes a new benchmark, NC-Bench, for NC pair prediction and introduces NCfold to address the data sparsity issue inherent in the NC-Bench dataset. To the best of our knowledge, no ethical concerns arise beyond those commonly associated with research in this field.

## REPRODUCIBILITY STATEMENT

We have described the model architecture, training procedure, and evaluation protocols in detail within the paper to ensure reproducibility. Upon acceptance, we will make all source data and the full implementation publicly available.

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

## A    THE USE OF LARGE LANGUAGE MODELS

A large language model (LLM) was used in a strictly limited manner during the preparation of this manuscript, solely to assist with minor language editing tasks such as grammar checking and formatting. All ideas, methods, analyses, and scientific interpretations were entirely developed and written by the authors without the involvement of the LLM. The authors thoroughly reviewed and, when necessary, revised any text suggested by the LLM to ensure that the final manuscript fully represents their own original work.

## B    NC-BENCH BENCHMARK DETAILS

### B.1    DATASET STATISTICS

Table 3: Distribution of non-canonical (NC) base-pair edges and orientations in the NC-Bench dataset (925 RNA sequences with 6,708 annotated NC pairs).

| Pair-edge family | Count | Percentage | Average per sequence |
|---|---|---|---|
| WS | 513 | 7.65% | 0.55 |
| HS | 593 | 8.84% | 0.64 |
| SS | 515 | 7.68% | 0.56 |
| WH | 1,807 | 26.94% | 1.95 |
| SH | 1,241 | 18.50% | 1.34 |
| HH | 228 | 3.40% | 0.25 |
| HW | 926 | 13.80% | 1.00 |
| SW | 885 | 13.19% | 0.96 |
| **Total NC pairs** | **6,708** | **100%** | **7.25** |
| **Orientation** | **Count** | **Percentage** | **Average per sequence** |
| cis | 2,753 | 41.04% | 379.62 |
| trans | 3,955 | 58.96% | 545.38 |
| **Total NC pairs** | **6,708** | **100%** | **7.25** |

Table 3 reports the distribution of non-canonical base-pair edges and orientations in the NC-Bench dataset (925 RNA sequences with 6,708 annotated NC pairs). Following the Leontis-Westhof scheme (Leontis & Westhof, 2001), we summarize eight pair-edge families (W/H/S denote Watson-Crick, Hoogsteen, and Sugar edges, respectively) and two orientations (cis/trans). The WH family is the most prevalent (26.94%, 1.95 pairs per sequence on average), followed by SH (18.50%), HW (13.80%), and SW (13.19%), whereas HH is rare (3.40%; 0.25 per sequence). Overall, sequences contain on average 7.25 NC pairs. For orientations, trans dominates (58.96%) over cis (41.04%). Based on counts, the correct per-sequence averages should be 2.98 (cis) and 4.28 (trans), summing to 7.25. Note that the screenshot shows 379.62 and 545.38 under Average per sequence; these values equal the percentage times the number of sequences (925) and therefore do not represent per-sequence averages. We will correct this column to avoid confusion. These statistics reveal substantial class imbalance across edge families and a strong preference for the trans orientation, which should be considered in model training (e.g., class weighting or resampling) and in reporting both micro- and macro-averaged evaluation metrics.

### B.2    EVALUATION METRICS

For performance evaluation of predicted NC edges and orientations, we use MCC, ACC, P, R, and F1 to assess the quality of base pair prediction. These metrics are computed based on the confusion matrix of classification task, where true positive (TP) is the number of correctly predicted edges/orientations, false positive (FP) is the number of incorrectly predicted edges/orientations, false negative is the number of failure predictions of edges/orientations, and true positive (TN) is correctly

predicted non-edge/non-pair, respectively. Based on these, metrics are defined as follows:

$$\text{MCC} = \frac{\text{TP} \times \text{TN} - \text{FP} \times \text{FN}}{\sqrt{\text{TP} + \text{FP}}\sqrt{\text{TN} + \text{FN}}\sqrt{\text{FP} + \text{TN}}\sqrt{\text{TP} + \text{FN}}},$$

$$\text{ACC} = \frac{\text{TP} + \text{TN}}{\text{TP} + \text{TN} + \text{FP} + \text{FN}},$$

$$\text{P} = \frac{\text{TP}}{\text{TP} + \text{FP}},$$

$$\text{R} = \frac{\text{TP}}{\text{TP} + \text{FN}},$$

$$\text{F1} = \frac{2 \times \text{P} \times \text{R}}{\text{P} + \text{R}}.$$

## C  MORE DETAILS OF BASELINES

In this section, we make a detailed description of baseline models that are used in our experiments, including traditional machine learning methods and RFMs.

### C.1  TRADITIONAL MACHINE LEARNING MODELS

We implement several classical machine learning models as baselines for our experiments. All models are configured to produce probabilistic outputs and are implemented using Scikit-learn.

- **Random Forest (RF)**: 400 trees with unlimited depth, balanced subsampling for class weighting, and parallelized computation.
- **Gradient Boosting**: 300 trees with learning rate 0.1.
- **XGBoost**: 300 trees with maximum depth 6, learning rate 0.1, and multi-class softmax objective.
- **Stochastic Gradient Descent (SGD)**: Logistic regression variant with log loss and maximum 2000 iterations.
- **Logistic Regression**: L2 regularization with balanced class weights and maximum 2000 iterations.
- **Support Vector Machine (SVM)**: RBF kernel with C=1.0 and probability estimates enabled.
- **K-Nearest Neighbors (KNN)**: 15 neighbors with distance-weighted voting.

### C.2  RNA FOUNDATION MODELS

As for RFMs, we construct the NC pair prediction model by simply employing the sequence embeddings generated by each RFM, followed by a learnable linear layer. The output and its inner production of this model construct the edge and orientation prediction, respectively.

## D  MORE EXPERIMENTS

### D.1  DETAILS OF SELECTION OF RFM EMBEDDINGS

Figure 5 summarizes the selection frequency of different RNA foundation models (RFMs) during the construction of Representative Embedding Fusion (REF) under various top-k settings. When only the top-1 model is selected, UTR-LM (Chu et al., 2024) dominates exclusively, being chosen in all 904 sequences. As the top-k increases, additional RFMs start to contribute: under top-2, RiNALMo (Penić et al., 2025) and RNA-FM (Chen et al., 2022) appear with notable frequencies (599 and 248, respectively); under top-3, four RFMs (UTR-LM, RiNALMo, RNA-FM, and strucRFM (Zhu et al., 2025a)) are all frequently selected, and by top-4 nearly all candidate RFMs are involved. This trend indicates that while UTR-LM embeddings are consistently strong and widely

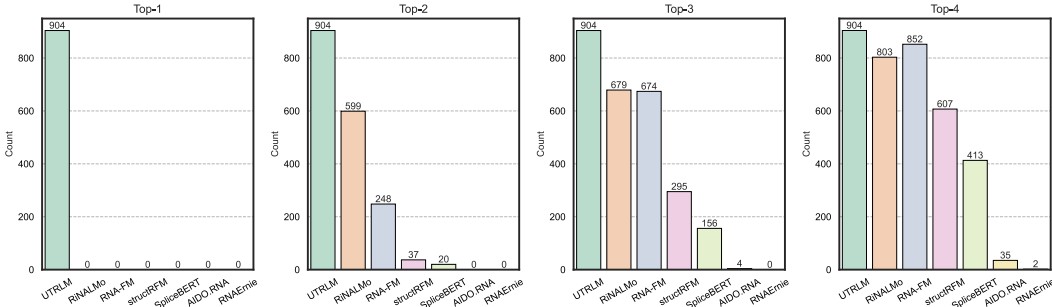

Figure 5: Statistics of Representative Embedding Fusion (REF) frequency of each RNA foundation model (RFM) under different settings of top-k.

preferred, integrating multiple RFMs becomes increasingly beneficial when the system is allowed to select more candidates. The diverse composition of REF at higher top-k settings highlights the complementary strengths among RFMs.

## D.2 Hyperparameter Selection

Table 4: Ablation study of NCfold under the different settings of transformer layers.

| Layers | Pair Edge | | | | | Orientation | | | | |
|---|---|---|---|---|---|---|---|---|---|---|
| | MCC | ACC | P | R | F1 | MCC | ACC | P | R | F1 |
| 2 | 0.131 | 0.602 | 0.254 | 0.345 | 0.271 | 0.214 | 0.927 | 0.421 | 0.528 | 0.433 |
| 4 | 0.170 | **0.614** | 0.319 | 0.359 | 0.305 | **0.285** | 0.953 | 0.461 | **0.546** | **0.472** |
| 6 | **0.219** | 0.603 | **0.370** | 0.376 | **0.336** | 0.265 | 0.948 | 0.462 | 0.520 | 0.466 |
| 8 | 0.194 | 0.583 | 0.355 | **0.377** | 0.330 | 0.268 | 0.960 | 0.467 | 0.508 | 0.470 |
| 10 | 0.206 | 0.540 | 0.357 | 0.371 | 0.317 | 0.270 | **0.955** | **0.474** | 0.506 | 0.471 |

Table 5: Ablation study of NCfold under the different settings of training batch size.

| Batch size | Pair Edge | | | | | Orientation | | | | |
|---|---|---|---|---|---|---|---|---|---|---|
| | MCC | ACC | P | R | F1 | MCC | ACC | P | R | F1 |
| 1 | 0.194 | 0.604 | 0.347 | 0.366 | 0.321 | 0.256 | 0.935 | **0.458** | 0.516 | 0.456 |
| 4 | **0.207** | 0.637 | **0.370** | **0.381** | **0.338** | **0.358** | **0.975** | 0.457 | **0.551** | **0.480** |
| 8 | 0.161 | 0.643 | 0.341 | 0.368 | 0.316 | 0.303 | 0.958 | 0.406 | 0.551 | 0.431 |
| 16 | 0.149 | 0.618 | 0.337 | 0.325 | 0.291 | 0.239 | 0.952 | 0.400 | 0.523 | 0.415 |
| 32 | 0.067 | 0.631 | 0.251 | 0.311 | 0.265 | 0.207 | 0.898 | 0.379 | 0.535 | 0.380 |
| 64 | 0.061 | **0.666** | 0.273 | 0.335 | 0.296 | 0.212 | 0.872 | 0.379 | 0.530 | 0.381 |

Tables 4 and 5 present ablation studies on the effects of transformer depth and training batch size in NCfold. As shown in Table 4, increasing the number of layers from 2 to 6 gradually improves performance on Pair Edge prediction, reaching the highest MCC (0.219) and F1 (0.336) at 6 layers, while further increasing to 8 or 10 layers leads to slight degradation, likely due to overfitting or optimization difficulty. Orientation prediction is less sensitive to depth, with relatively stable performance across 4-10 layers and the best F1 (0.472) at 4 layers. Table 5 shows that batch size has a stronger effect: a small-to-moderate batch size of 4 yields the best performance on both tasks (Pair Edge: MCC 0.207, F1 0.338; Orientation: MCC 0.358, F1 0.480), whereas larger batch sizes (> 16) noticeably reduce MCC and F1 despite slightly higher accuracy. These results suggest that NCfold

benefits from a moderate model capacity and small batch training, while deeper or larger-batch configurations harm its generalization.

### D.3 ZERO-SHOT ASSESSMENT

Table 6: Zero-shot evaluation of RNA secondary structure methods for NC pairing prediction. As existing methods only predict pairing status, namely "non-pair" and "pair". We convert the predicted orientations (i.e., "trans" and "cis") of NCfold to the "paired" category for reference comparison. The best results are in **bold**.

| Model | Pairing Status | | | | |
|-------|------|------|------|------|------|
| | MCC | ACC | P | R | F1 |
| BPfold | 0.799 | 0.965 | 0.885 | 0.021 | 0.033 |
| UFold | 0.625 | 0.965 | 0.752 | 0.027 | 0.047 |
| MXfold2 | **0.984** | 0.965 | **0.984** | 0.000 | 0.000 |
| SPOT-RNA | 0.573 | 0.965 | 0.757 | 0.022 | 0.039 |
| NCfold | 0.463 | **0.968** | 0.489 | **0.431** | **0.440** |

Table 6 shows the zero-shot performance of several RNA secondary structure predictions on NC pairing status classification. While all baseline methods (BPFold (Zhu et al., 2025b), UFold (Fu et al., 2022), MXfold2 (Sato et al., 2021), SPOT-RNA (Singh et al., 2019)) achieve high overall accuracy ( 0.965), they almost never predict any paired positions, leading to extremely low recall (¡0.03) and near-zero F1 scores. For example, MXfold2 attains the highest MCC (0.984) and precision (0.984) but a recall of 0, indicating that it predicts all residues as non-paired. In contrast, NCfold achieves substantially higher recall (0.431) and F1 (0.440), demonstrating its ability to actually detect NC pairs, though with a lower MCC (0.463) due to increased false positives. These results highlight the limitations of canonical structure predictors for NC pairing and the necessity of specialized models such as NCfold.

### D.4 THE MACRO-AVERAGED ACCURACY OF THE PER-CLASS RESULTS.

Table 7: The macro-averaged accuracy of per-class results of NCfold.

| Model | Pair-Edge | | | | Orientation | | |
|-------|----------|---|---|---|------------|---|---|
| | Non-edge | W | H | S | Non-pair | trans | cis |
| NCfold | 0.200 | 0.837 | 0.192 | 0.224 | 0.978 | 0.006 | 0.567 |

As Tables 7 demonstrated, NCfold achieves highly macro-averaged accuracy for Watson-Crick edge of 0.837 than other edge types (including Non-edge, Hoogsteen, Sugar). As for base-to-base orientation, due to the scarcity of no-canonical base pairs, NCfold achieves higher accuracy in Non-pair of 0.978, relatively good performance on cis orientations, and extremely low accuracy of 0.006 for trans orientation.

### D.5 THE DETAILED 4-FOLD PERFORMANCE OF NCFOLD.

Table 8: The detailed 4-fold performance of NCfold. We show the performances of different fold of NCfold, together with the variance and mean value.

| Fold | Pair Edge | | | | | Orientation | | | | |
|------|------|------|------|------|------|------|------|------|------|------|
| | MCC | ACC | P | R | F1 | MCC | ACC | P | R | F1 |
| 1 | 0.162 | 0.604 | 0.336 | 0.372 | 0.309 | 0.286 | 0.954 | 0.415 | 0.562 | 0.439 |
| 2 | 0.198 | 0.643 | 0.371 | 0.368 | 0.337 | 0.322 | 0.975 | 0.442 | 0.537 | 0.467 |
| 3 | 0.223 | 0.651 | 0.387 | 0.381 | 0.352 | 0.345 | 0.974 | 0.447 | 0.556 | 0.473 |
| 4 | 0.193 | 0.631 | 0.379 | 0.370 | 0.340 | 0.347 | 0.975 | 0.449 | 0.554 | 0.475 |
| mean±var | 0.194 ± 0.001 | 0.632 ± 0.000 | 0.368 ± 0.001 | 0.373 ± 0.000 | 0.335 ± 0.000 | 0.325 ± 0.001 | 0.969 ± 0.000 | 0.438 ± 0.000 | 0.552 ± 0.000 | 0.463 ± 0.000 |
| NCfold | 0.193 | 0.631 | 0.379 | 0.370 | 0.340 | 0.347 | 0.975 | 0.449 | 0.554 | 0.475 |

## D.6 More Visualization Results

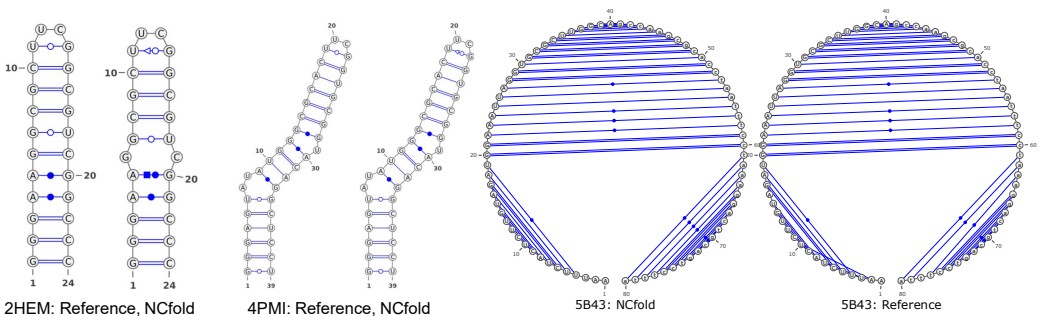

Figure 6: More visualization of NC edges (Watson-Crick ○/●, Hoogsteen □/■, and Sugar △/▲) and corresponding orientations (trans/cis) from reference structures and predictions by NCfold.

# E Discussion

In this paper, we introduce NC-Bench, the first dedicated benchmark for predicting non-canonical (NC) base pairs in RNA structures. NC-Bench includes 925 sequences with 6,708 annotated edges and orientations, along with Isoscore-based evaluation for RFMs, providing a comprehensive standard for model assessment and development. Building upon this benchmark, we propose NCfold, a transformer-based neural network that incorporates RFMs as priors for base-to-base interactions via a biased self-attention mechanism. This design enables joint modeling of sequential and structural features, thereby mitigating issues related to data scarcity. Extensive experiments demonstrate that NCfold outperforms both traditional machine learning methods and RFM-based approaches.

Despite these contributions, NC-Bench and NCfold have several limitations that suggest directions for future work. First, the current scale of NC-Bench remains limited. It could be expanded by extracting additional NC annotations from predicted RNA tertiary structures, which could serve as augmented training data. Moreover, such computationally derived annotations could be further refined and filtered using more detailed structural rules. Moreover, NC-Bench can further be extended with more sophisticated tertiary contacts (such as base stacking) and multi-base interactions, which challenges the method development for predicting higher-order interactions and dealing with data scarsity even rarer than non-canonical base pairs. Second, NCfold currently predicts edges and orientations directly from the model outputs. Incorporating prior knowledge-based constraints during post-processing could further improve results. For instance, as implemented in BPfold Zhu et al. (2025b), sharp loops or isolated base pairs are not allowed in predictions. Also, future work may enhance the REF-weighted self-attention processing by replacing the standard convolution with row- and column-wise attention for more global view. Third, NCfold relies on fully supervised training, which is challenging given the limited dataset size. Future work could explore few-shot learning techniques to better capture the distribution of NC pairs and enhance prediction performance.

