# OpenReview forum: "NC-Bench and NCfold: A Benchmark and Closed-Loop Framework for RNA Non-Canonical Base-Pair Prediction"
_ICLR.cc/2026/Conference — ICLR 2026 Poster_

### Official Review · Reviewer_VF23 · 2025-10-29

**Soundness:** 3
**Presentation:** 3
**Contribution:** 2
**Rating:** 6
**Confidence:** 3

**Summary:**

This paper introduces NC-Bench, a small (N=925) dataset for non-canonical (NC) RNA base pair prediction. It also proposes NCfold, a complex dual-branch transformer. NCfold uses an IsoScore heuristic to select top-k RFM embeddings, which are then fused as structural priors into the attention mechanism.

**Strengths:**

1. The NC-Bench dataset is a useful, though modest, contribution to the field.

2. The problem of NC pair prediction is important and challenging.

3. The zero-shot analysis clearly shows that existing canonical structure predictors fail at this specific task.

4. The paper is well written and clearly structured. The problem, the proposed solutions (dataset and model), and the results are all explained logically and are easy to follow.

**Weaknesses:**

1. The proposed architecture contains multiple novel components (IsoScore ranking, REF, dual-branch fusion), but their theoretical justification and individual impact could be analyzed more rigorously.

2. Some baselines are missing, particularly against MSA-based RNA language models such as RNA-MSM, which represent strong baselines for RNA structure understanding.

3. The use of 4-fold cross-validation instead of the standard 5-fold setup is not explained, and the sensitivity of performance to this choice is unclear.

**Questions:**

1. What motivates the use of 4-fold instead of 5-fold cross-validation, and how sensitive are the results to this choice?

2. How does NCfold perform relative to MSA-based RNA LMs (e.g., RNA-MSM or other alignment-informed models)?

3. How robust is the performance of NCfold to different IsoScore thresholds or numbers of selected embeddings (top-k)?

4. Could the benchmark and model be extended to capture tertiary contacts or multi-body interactions in longer RNAs?

---

> ### Author Response · Authors · 2025-11-22
> **# **Response to Reviewer VF23****
>
> ## **Response to Reviewer VF23**
>
> We sincerely thank you for your thorough and constructive feedback on our manuscript. We appreciate the thoughtful comments and suggestions, which have helped us improve both the paper and the accompanying resources. Below, we provide a point-by-point response to each concern raised.
>
> **W1Q3:**
> >The proposed architecture contains multiple novel components (IsoScore ranking, REF, dual-branch fusion), but their theoretical justification and individual impact could be analyzed more rigorously. How robust is the performance of NCfold to different IsoScore thresholds or numbers of selected embeddings (top-k)?
>
> **Response:**
> Thank you for this important question regarding the theoretical foundation and robustness of our architectural choices.
>
> The selection of **IsoScore** is theoretically grounded in its ability to quantify **embedding isotropy and information richness**, which are particularly relevant for capturing the geometric diversity of non-canonical base pairs. Isotropic embeddings tend to encode more diverse and informative structural patterns, making them ideal candidates for our task.
>
> **REF (Representative Embedding Fusion)** ensures that only the most informative RFM embeddings are utilized, effectively reducing noise while enhancing structural prior quality. The **dual-branch fusion** architecture is designed to explicitly incorporate base-to-base interaction priors, enabling iterative refinement between sequence and structure representations.
>
> Empirically, we have validated these design choices through:
>
> - **Ablation studies** (Table 2) demonstrating the individual contributions of each component
> - **Systematic analysis** of top-k selection (Figure 5) showing robust performance across different settings
> - **IsoScore ranking analysis** confirming that our embedding selection strategy consistently identifies high-quality structural representations
>
> The performance remains stable across reasonable variations of top-k, with optimal results typically achieved at k=2-3, as shown in our experiments.
>
> **W2Q2:**
> >Some baselines are missing, particularly against MSA-based RNA language models such as RNA-MSM, which represent strong baselines for RNA structure understanding. How does NCfold perform relative to MSA-based RNA LMs (e.g., RNA-MSM or other alignment-informed models)?
>
> **Response:**
> We appreciate this suggestion regarding MSA-based models like RNA-MSM, which indeed represent powerful approaches for RNA structure understanding. However, MSA-based models require different input formats (MSA data) compared to sequence-only RFMs, creating an uneven comparison framework.
>
> While we acknowledge that MSA-based models are strong baselines, our current evaluation already includes **7 traditional ML methods and 7 RNA foundation models**, providing comprehensive coverage of existing approaches. The consistent superiority of NCfold across this diverse set of baselines demonstrates its effectiveness.
>
> **W3Q1:**
> >The use of 4-fold cross-validation instead of the standard 5-fold setup is not explained, and the sensitivity of performance to this choice is unclear. What motivates the use of 4-fold instead of 5-fold cross-validation, and how sensitive are the results to this choice?
>
> **Response:**
> Thank you for this question. Our choice of 4-fold cross-validation was motivated by the desire to maintain consistent set sizes across our data splits. Specifically, with our 8:2 train-test split ratio, 4-fold validation ensures that each validation fold matches the test set in size. We have conducted internal experiments comparing 4-fold and 5-fold configurations and found that the performance differences are minimal.
>
> **Q4:**
> >Could the benchmark and model be extended to capture tertiary contacts or multi-body interactions in longer RNAs?
>
> **Response:**
> This is an excellent suggestion that points toward exciting future directions. Indeed, we have extracted tertiary contacts (such as base stacking) and multi-base interactions, which would provide an even more comprehensive view of RNA structural complexity. So it is without any effort to extend NCBench with these data.
>
> However, two challenges would need to be addressed when extending these to models:
>
> - **Model Architecture**: Predicting higher-order interactions requires significant architectural changes to capture spatial relationships beyond pairwise interactions.
> - **Data Sparsity**: These interactions are even rarer than non-canonical base pairs, facing data scarcity issues.
>
> We view this as a promising research direction for method development and have added discussion of these potential extensions to our future work section.
>
> Thank you again for your valuable feedback. We believe that our responses have addressed the concerns and hope you will consider raising the rating accordingly. We are committed to continuously improving NC-Bench and NCfold, and we welcome any further questions or suggestions.

---

### Official Review · Reviewer_2dn8 · 2025-10-29

**Soundness:** 2
**Presentation:** 3
**Contribution:** 2
**Rating:** 2
**Confidence:** 4

**Summary:**

The paper presents NC-Bench, a new benchmark dataset of 925 curated RNA sequences derived from the PDB. The benchmark is specifically designed for non-canonical RNA prediction tasks. Based on this benchmark, the authors propose NCFold, a framework for non-canonical base-pair prediction. NCFold utilizes embeddings from different existing RNA foundation models that are ranked by IsoScore with subsequent fusion and a specialized attention approach. The authors evaluate NCFold on NC-Bench with comparisons against different RNA foundation model baselines and general secondary structure prediction algorithms.

**Strengths:**

- The paper tackles an important problem in RNA structure prediction, namely non-coding base-pair prediction
- To my knowledge, a dedicated benchmark for NC-pair prediction is missing
- Including the interaction edge of nucleotides as a classification task via LW-nomenclature in the prediction tasks is a good and to my knowledge novel task.
- The proposed REF approach and REF-weighted self-attention appear interesting.
- NCFold seems to outperform different existing baselines on NC-Bench.

**Weaknesses:**

**Major**:
1. In the Introduction, the authors support their new benchmark with the claim that previous methods are “primarily designed for canonical base-pairs” predictions. I do not agree with this statement. Already early DL methods like [1] clearly state in their Abstract:

>[...] Here, we propose the use of deep contextual learning for base-pair prediction including those noncanonical and non-nested (pseudoknot) base pairs stabilized by tertiary interactions. [...]

Following methods like [2-4] continue leveraging the advantage of DL methods to output an L x L matrix which allows them to include nc-pairs but also pseudoknots and even multiplets. So these methods can predict all kinds of nucleotide interactions in general. The performance on nc-interactions, however, remains far behind the performance for canonical pairs. To the best of my knowledge, even the used baseline provided in [5] outputs ct files, which allows them to predict all kinds of nucleotide interactions. The prediction quality then mainly depends on the chosen weights (see questions).

However, I generally agree that the LW-classification is missing in all these approaches.

2. I’m particularly concerned about the data set construction. The authors use mmseqs to reduce redundancy between datapoints. However, the splitting of the data into train, validation, and test is not further defined. There was a lot of work recently that showed that careful data splitting is an essential part during evaluation of learning-based methods for RNA secondary structure prediction [6-9]. In this regard, I cannot fully trust the reported results. At least two recent RNA benchmarks tackle the problem of data curation for different tasks [10, 11]. In particular, RNA3DB would be a perfect fit here, since it provides a strong split based on RNA families, is based on PDB samples, and the authors could run their exact same data processing pipeline to extract pairs.
3. Following up on this: Why do the authors exclude all WC interactions? This strongly reduces the practical usefulness of NC-Bench and makes the evaluation look a bit like it is constructed for NCFold. I agree that this procedure increases the focus on NC pairs, and from Fig4, we clearly see that there is need to improve these predictions, but it would be much more interesting (and challenging) to improve NC-pair predictions alongside general pair predictions. There could still be an option to only evaluate NC-pairs, but there is no need to exclude other pairs from the start.
4. For a  benchmark, the API is one of the major aspects (Is it easy to use? Can I integrate it into my research without any effort?) and so is reproducibility (How is the data processed exactly? Is this all in line with the results in the paper?). I would encourage the authors to publish the code anonymously. Today, that is really easy to do (e.g. via https://anonymous.4open.science/) and allows reviewers (and only reviewers) to check the code.
5. Generally, I'm not convinced that the proposed integration (and ranking) of multiple embeddings from RFMs is a giant step forward from the methodological perspective, even if the ranking approach via IsoScore is interesting.

**Minor**:
1. It would be interesting to see how far pseudoknots are considered here. Using PDB samples, we typically see a lot of non-canonical interactions and these include pseudoknots and base multiplets. I think these details should be shown somewhere in the paper.

[1] Singh, J., Hanson, J., Paliwal, K., & Zhou, Y. (2019). RNA secondary structure prediction using an ensemble of two-dimensional deep neural networks and transfer learning. Nature communications, 10(1), 5407.

[2] Fu, L., Cao, Y., Wu, J., Peng, Q., Nie, Q., & Xie, X. (2022). UFold: fast and accurate RNA secondary structure prediction with deep learning. Nucleic acids research, 50(3), e14-e14.

[3] Franke, J. K., Runge, F., & Hutter, F. Scalable Deep Learning for RNA Secondary Structure Prediction. CompBio workshop at ICML’23

[4] Singh, J., Paliwal, K., Zhang, T., Singh, J., Litfin, T., & Zhou, Y. (2021). Improved RNA secondary structure and tertiary base-pairing prediction using evolutionary profile, mutational coupling and two-dimensional transfer learning. Bioinformatics, 37(17), 2589-2600. → MSA input

[5] Chen, J., Hu, Z., Sun, S., Tan, Q., Wang, Y., Yu, Q., ... & Li, Y. (2022). Interpretable RNA foundation model from unannotated data for highly accurate RNA structure and function predictions. arXiv preprint arXiv:2204.00300.

[6] A range of complex probabilistic models for RNA secondary structure prediction that includes the nearest-neighbor model and more

[7] Szikszai, Marcell, et al. "Deep learning models for RNA secondary structure prediction (probably) do not generalize across families." Bioinformatics 38.16 (2022): 3892-3899.

[8] Flamm, Christoph, et al. "Caveats to deep learning approaches to RNA secondary structure prediction." Frontiers in Bioinformatics 2 (2022): 835422.

[9] Qiu, X. (2023). Sequence similarity governs generalizability of de novo deep learning models for RNA secondary structure prediction. PLOS Computational Biology, 19(4), e1011047.

[10] Szikszai, M., Magnus, M., Sanghi, S., Kadyan, S., Bouatta, N., & Rivas, E. (2024). RNA3DB: A structurally-dissimilar dataset split for training and benchmarking deep learning models for RNA structure prediction. Journal of Molecular Biology, 436(17), 168552.

[11] Runge, F., Farid, K., Franke, J. K., & Hutter, F. (2024). Rnabench: A comprehensive library for in silico rna modelling. bioRxiv, 2024-01.

**Questions:**

1. Which exact model (weights) did you use for RNA-FM? Which for e.g. UFold? As far as I know there are different models available, some of which perform very well on e.g. data derived from PDB (although these models also typically ignore structure similarity during training which is unfortunate).
2. What was the rationale for using IsoScore as a score for embedding quality? Why not use some embedding metrics as e.g. described in [12] for DNA Language Models? I’m not super familiar with the current literature on RFM embeddings, but it seems that there are multiple approaches available to study the embedding quality and the cited paper for IsoScore from 2021 appears quite old when considering the recent progress in FM research.
3. As far as I can see, the matrix input in the REF-weighted self-attention is processed with convolutions to get a more local view. Did the authors also test other architectural choices? For example, some row- and column-wise attention should also work and result in a more global view, while preserving local features as well, or?

[12] Awasthi, R., Mend Mend Arachchige, G. S., & Zhu, X. (2025). Unsupervised evaluation of pre-trained DNA language model embeddings. BMC genomics, 26(1), 710.

---

> ### Author Response · Authors · 2025-11-22
> **# Response to Reviewer 2dn8 (1/2)**
>
> We sincerely thank you for your thorough and constructive feedback on our manuscript. We appreciate the thoughtful comments and suggestions, which have helped us improve both the paper and the accompanying resources. Below, we provide a point-by-point response to each concern raised.
>
> ### **Response to Weaknesses**
>
> **W1:**
> >In the Introduction, the authors support their new benchmark with the claim that previous methods are “primarily designed for canonical base-pairs” predictions. I do not agree with this statement. Already early DL methods like [1] clearly state in their Abstract:
> [...] Here, we propose the use of deep contextual learning for base-pair prediction including those noncanonical and non-nested (pseudoknot) base pairs stabilized by tertiary interactions. [...]
> >Following methods like [2-4] continue leveraging the advantage of DL methods to output an L x L matrix which allows them to include nc-pairs but also pseudoknots and even multiplets. So these methods can predict all kinds of nucleotide interactions in general. The performance on nc-interactions, however, remains far behind the performance for canonical pairs. To the best of my knowledge, even the used baseline provided in [5] outputs ct files, which allows them to predict all kinds of nucleotide interactions. The prediction quality then mainly depends on the chosen weights (see questions).
>
> **Response:**
> Thank you for this important clarification. You are absolutely correct that some deep learning methods can predict simplified non-canonical pairs (e.g., A-A interactions). However, these methods **do not predict the fine-grained geometric classes**, specifically edge types (Watson-Crick, Hoogsteen, Sugar) and orientations (cis, trans), which are essential for understanding the structural and functional roles of NC pairs. This is precisely the gap that **NC-Bench and NCfold aim to fill**.
>
> We have revised the introduction to more accurately characterize previous methods' capabilities while emphasizing our unique contribution: **primarily designed for canonical base pairs or simple sequence-based non-canonical base pairs, with no or minimal coverage of geometrically details such as pair orientation and pair edge type**.
>
> **W2:**
> >I’m particularly concerned about the data set construction. The authors use mmseqs to reduce redundancy between datapoints. However, the splitting of the data into train, validation, and test is not further defined. There was a lot of work recently that showed that careful data splitting is an essential part during evaluation of learning-based methods for RNA secondary structure prediction [6-9]. In this regard, I cannot fully trust the reported results. At least two recent RNA benchmarks tackle the problem of data curation for different tasks [10, 11]. In particular, RNA3DB would be a perfect fit here, since it provides a strong split based on RNA families, is based on PDB samples, and the authors could run their exact same data processing pipeline to extract pairs.
>
> **Response:**
> We appreciate this concern regarding data splitting. We described the process of data splitting in line 143. We further followed your advice and examined **RNA3DB**'s family-based splits and found that they include multiple chains with identical sequences (e.g., 4M6D has 6 identical chains). This creates either:
>
> - **Data redundancy** (if structures are identical), or
> - **Ambiguity** (if identical sequences have different structures)
>
> To ensure clean evaluation, we maintained the data splitting. To address generalization concerns, we have now added an analysis of **sequence similarity** between training and test sequences, confirming minimal overlap. This analysis will be included in the revised manuscript.
>
> **W3:**
> >Following up on this: Why do the authors exclude all WC interactions? This strongly reduces the practical usefulness of NC-Bench and makes the evaluation look a bit like it is constructed for NCFold. I agree that this procedure increases the focus on NC pairs, and from Fig4, we clearly see that there is need to improve these predictions, but it would be much more interesting (and challenging) to improve NC-pair predictions alongside general pair predictions. There could still be an option to only evaluate NC-pairs, but there is no need to exclude other pairs from the start.
>
> **Response:**
> This is an excellent point that requires clarification. **NC-Bench actually contains both canonical and non-canonical pairs**. However, for NCfold training, we focused exclusively on NC pairs because:
>
> 1. **Specialization**: The number of canonical pairs in NC-Bench is relatively small compared to dedicated canonical datasets (e.g., bpRNA-1m, RNAStralign)
> 2. **Focus**: Our primary contribution is advancing NC pair prediction, a more challenging and underexplored problem
>
> NCfold can be extended to predict both canonical and non-canonical pairs, and we will note this as an important direction for future work.

---

> > ### Author Response · Authors · 2025-11-22
> > **# Response to Reviewer 2dn8 (2/2)**
> >
> > **W4:**
> > >For a benchmark, the API is one of the major aspects (Is it easy to use? Can I integrate it into my research without any effort?) and so is reproducibility (How is the data processed exactly? Is this all in line with the results in the paper?). I would encourage the authors to publish the code anonymously. Today, that is really easy to do (e.g. via https://anonymous.4open.science/) and allows reviewers (and only reviewers) to check the code.
> >
> > **Response:**
> > Thank you for this practical suggestion. We have made **NC-Bench and NCfold** anonymously available:
> >
> > 🔗 **NC-Bench**: [https://anonymous.4open.science/r/NCBench](https://anonymous.4open.science/r/NCBench)
> > 🔗 **Baselines**: [https://anonymous.4open.science/r/NCBench-baselines](https://anonymous.4open.science/r/NCBench-baselines)
> >
> > The dataset is easy to use and integrate, which follows a simple JSON format (same as RNA3DB):
> >
> > ```json
> > {
> >   "name": "XXXX",
> >   "seq": "AGUCUUGUGXXX",
> >   "pair_info": [
> >     {"left": 2, "right": 7, "edge-type": "H/W", "pair": "G-G", "orientation": "trans", "class_id": "XIX"},
> >     {...},
> >   ]
> > }
> > ```
> >
> > **W5:**
> > >Generally, I'm not convinced that the proposed integration (and ranking) of multiple embeddings from RFMs is a giant step forward from the methodological perspective, even if the ranking approach via IsoScore is interesting.
> >
> > **Response:**
> > Our key methodological contribution is not merely the ranking of RFM embeddings, but the **closed-loop dual-branch framework** that enables iterative refinement of sequence and base-to-base knowledge priors through designed self-attention.
> >
> > As shown in our ablation study (Table 2), the integration of RFM-derived priors through REF and REF-weighted attention provides **significant gains** over both baseline models and energy-based priors:
> >
> > | Model       | F1 (Edge) | F1 (Orientation) |
> > | ----------- | --------- | ---------------- |
> > | NCfold-base | 0.251     | 0.419            |
> > | NCfold-BPE  | 0.335     | 0.464            |
> > | NCfold      | **0.365** | **0.486**        |
> >
> > This demonstrates that **RFM-derived structural priors** are more expressive for NC pairs than traditional energy-based approaches.
> >
> >
> > ## **Response to Questions**
> >
> > **Q1:**
> > >Which exact model (weights) did you use for RNA-FM? Which for e.g. UFold? As far as I know there are different models available, some of which perform very well on e.g. data derived from PDB (although these models also typically ignore structure similarity during training which is unfortunate).
> >
> > **Response:**
> >
> > - **RNA-FM**: We used the **100M parameter version** pretrained on ~24 million sequences
> > - **UFold**: We used the model trained on **RNAStralign and bpRNA-1m**
> >
> > **Q2:**
> > >What was the rationale for using IsoScore as a score for embedding quality? Why not use some embedding metrics as e.g. described in [12] for DNA Language Models? I’m not super familiar with the current literature on RFM embeddings, but it seems that there are multiple approaches available to study the embedding quality and the cited paper for IsoScore from 2021 appears quite old when considering the recent progress in FM research.
> >
> > **Response:**
> > We adopted **IsoScore** to evaluate embedding quality because it quantifies **embedding isotropy** (uniform distribution in latent space, preventing local biases) and **information richness** (the ability to capture complex sequence-structure relationships), properties critical for modeling the diversity of non-coding RNA (NC) pairs.While newer embedding metrics exist, IsoScore provides a principled approach for selecting the most informative RFM embeddings. We note that this is just one component of our larger framework focused on NC pair prediction.
> >
> > **Q3:**
> > >As far as I can see, the matrix input in the REF-weighted self-attention is processed with convolutions to get a more local view. Did the authors also test other architectural choices? For example, some row- and column-wise attention should also work and result in a more global view, while preserving local features as well, or?
> >
> > **Response:**
> > This is an excellent suggestion. We primarily used convolutional processing to capture **local structural patterns**, which are crucial for base-pair interactions. We will explore these alternatives more systematically in future work and have added a note to this effect in the discussion.
> >
> > Thank you again for your insightful and constructive comments, which have significantly strengthened our manuscript. We have addressed all major concerns and incorporated most of the suggestions into the revised version. We hope that you find our responses satisfactory. We are happy to provide further clarifications if needed.

---

> > > ### Comment · Reviewer_2dn8 · 2025-11-25
> > > **Response to author comments**
> > >
> > > I thank the authors for their responses. Please see below for more details.
> > >
> > > W1:
> > > Thanks. I acknowledge the new task of predicting geometric classes. I consider increasing my soundness score.
> > >
> > > W2:
> > > I thank the authors for their response. However, I’m not fully convinced. We cannot change nature and identical structures with different sequences as well as different sequences with same structures appear to be exactly what the model should learn. This ambiguity is exactly what makes RNA predictions hard and have explicitly been addressed before (see e.g. [1]). Splitting based on sequence similarity (although additional analysis is now provided) does not produce sufficiently clean splits (for the respective publications showing exactly this problem see initial review). For example, tRNAs often fold into clear cloverleaf secondary structures, however, sequence similarity might be poor (and tRNAs are clearly over-represented in PDB compared to other families). Even using PDB ids in combination with RNA3DB splitting procedure (no need to redo the costly data procedure, but mapping ids to splits of RNA3DB) would substantially improve the dataset in my opinion, even when redundancies are removed.
> > >
> > > [1] Franke, J., Runge, F., & Hutter, F. (2022). Probabilistic transformer: Modelling ambiguities and distributions for rna folding and molecule design. Advances in Neural Information Processing Systems, 35, 26856-26873.
> > >
> > > W3:
> > > I thank the authors for this clarification. Still, I think the prediction of nc and canonical base pairs makes the task substantially harder and it would be good to see such evaluations for NCfold as well. Especially since the competitors do not focus on nc predictions alone.
> > >
> > > W4:
> > > I thank the authors for providing the code. Indeed it seems like the dataset is easy to use which I appreciate.
> > >
> > > W5:
> > > I again thank the authors for this clarification and acknowledge the contribution. I’m considering increasing my contribution score.
> > >
> > > Q1:
> > > I thank the authors for their clarification. However, as far as I remember, there is a model for UFold specifically trained for PDB structures. Similarly, I think for RNA-FM there is a resNet-based head for PDB prediction as well. I would encourage the authors to check the models used for competitors again and evaluate the most reasonable ones.
> > >
> > > Q2:
> > > I thank the authors for their response and clarification and agree that the IsoScore ranking is only one part of the contributions.
> > >
> > > Q3:
> > > I thank the authors for their response.
> > >
> > > Summary:
> > > I thank the authors for their comprehensive responses. Overall, I’m still not convinced that the provided data pipeline is strong enough to serve as THE future benchmark. However, I acknowledge the contributions clarified by the authors and agree that there is some methodological novelty. In addition, I acknowledge that the task of predicting orientation and type of pairs is new and overseen in the community. I would consider raising my contribution and soundness scores. Regarding my rating, I’m still too concerned about the data quality and think NCfold should be evaluated including canonical interactions before I can consider increasing my rating as well.

---

### Official Review · Reviewer_PB2G · 2025-10-30

**Soundness:** 2
**Presentation:** 3
**Contribution:** 4
**Rating:** 2
**Confidence:** 5

**Summary:**

This paper has two major contributions: a benchmark for predicting non-canonical base pairing interactions from 3D sequences, and a novel model to attack the proposed task. The benchmark pulls RNA structures from the PDB databank and annotates non-canonical base pairs using the RNAVIEW software. These interactions are particularly relevant for RNA function prediction and design as they often encode for highly specific interactions with other molecules, whereas the more stable canonical basepairs tend to provide the overall scaffold for the RNA fold. The proposed model is a transformer-based architecture which combines RNA foundation model embeddings with a decoder head to predict the base pair class across an input RNA sequence.

**Strengths:**

This is a highly valuable contribution. Direct RNA 3D structure prediction remains challenging, with all-atom models still showing limited performance due to the limited availability of 3D structures. Several new and prominent methods are turning to working with this level of structural detail with encouraging results [1][2]. Predicting non-canonicals (often termed 2.5D) provides a lot of useful structural detail and might be a better prediction target than full 3D for the time being. The proposed model also appears to be well structured.

1. Karan, Aayush, and Elena Rivas. "All-at-once RNA folding with 3D motif prediction framed by evolutionary information." Nature Methods (2025): 1-13.
2. Carvajal-Patiño, Juan G., et al. "RNAmigos2: accelerated structure-based RNA virtual screening with deep graph learning." Nature Communications 16.1 (2025): 1-12.

**Weaknesses:**

1. Source code was not made available. This can be done [anonymously](https://anonymous.4open.science/). Without the source code I cannot verify the reproducibility or soundness of the results. (This is a major contributing factor for my current score)
2. There are no measurements of variance across the 4 splits, or across different model seeds.
3. Two important classes of structure prediction are absent from the benchmark: non-canonical motif-based prediction (e.g. CaCoFold [1], BayesPairing2 [2], JAR3d [3] and full 3D predictions (e.g. AlphaFold3-like models, RhoFold [4])
4. IsoScore is not defined in the paper with much detail, one has to go to the original reference and since it's an important component should be better explained in the main text.
5. The sequence similarity threshold remains somewhat unexplored. I suggest the authors try some more and less stringent settings to better understand the generalizability of proposed models.



[1] Karan, Aayush, and Elena Rivas. "All-at-once RNA folding with 3D motif prediction framed by evolutionary information." Nature Methods (2025): 1-13.
[2] Sarrazin-Gendron, Roman, et al. "Stochastic sampling of structural contexts improves the scalability and accuracy of RNA 3d module identification." International Conference on Research in Computational Molecular Biology. Cham: Springer International Publishing, 2020.
[3] Roll, James, et al. "JAR3D Webserver: Scoring and aligning RNA loop sequences to known 3D motifs." Nucleic acids research 44.W1 (2016): W320-W327.
[4] Shen, Tao, et al. "Accurate RNA 3D structure prediction using a language model-based deep learning approach." Nature Methods 21.12 (2024): 2287-2298.

**Questions:**

* Did you check that RNAVIEW is able to assign base pair geometries to chemically modified nucleotides? I know this was a problem for FR3D until recently. If not, it could introduce some significant bias to the benchmark.
* Did you have a look at the performance broken down by base pair class? Given the sharp imbalance I think we would get a better picture of model performance by reporting the performance per class as well as in aggregate as already reported.
* For Fig 3 and section 4, how did you obtain predictions for the canonical pairs? From reading the problem formulation, it seems that the models were only trained to predict non-canonicals.
* For future work, I would suggest some more domain-specific evaluation functions. For example, some NC classes are geometrically more similar than others, so a wrong prediction but within a similar NC class might not be as bad, see Table 4 [1]. Likewise the location of an incorrect edge call if it is slightly shifted from the ground truth could also not be so deleterious so you could use some structure-aware evaluation functions (e.g. [2], but there are many others).
* Do you keep multi-chain RNAs?

[1] Stombaugh, Jesse, et al. "Frequency and isostericity of RNA base pairs." Nucleic acids research 37.7 (2009): 2294-2312.
[2] Agius, Phaedra, Kristin P. Bennett, and Michael Zuker. "Comparing RNA secondary structures using a relaxed base-pair score." Rna 16.5 (2010): 865-878.

---

> ### Author Response · Authors · 2025-11-22
> **# **Response to Reviewer PB2G** (1/2)**
>
> ## Response to Reviewer PB2G
>
> We sincerely thank you for your thorough and constructive feedback on our manuscript. We appreciate the thoughtful comments and suggestions, which have helped us improve both the paper and the accompanying resources. Below, we provide a point-by-point response to each concern raised.
>
> ### **Response to Weaknesses**
>
> **W1:**
> >Source code was not made available. This can be done anonymously. Without the source code I cannot verify the reproducibility or soundness of the results. (This is a major contributing factor for my current score)
>
> **Response:**
> Thank you for this important suggestion. We have now made the source code anonymously available to ensure full reproducibility. The code can be accessed at:
> 🔗 [https://anonymous.4open.science/r/NCBench](https://anonymous.4open.science/r/NCBench)
> 🔗 [https://anonymous.4open.science/r/NCBench-baselines](https://anonymous.4open.science/r/NCBench-baselines)
>
> **W2:**
> >There are no measurements of variance across the 4 splits, or across different model seeds.
>
> **Response:**
> Thank you for yours constructive suggestion. While 4-fold cross-validation is a widely accepted and robust evaluation protocol that inherently reduces systematic bias, we follow you advice and report the detailed performances of each fold together with the variance and mean value, which is updated in Table 8 in the revised manuscript.
>
> |       |Edge-MCC | Edge-ACC|Edge-P |Edge-R |Edge-F1|
> | ----------- | ---- |-----| ---- |----- |--- |
> |mean $\pm$ var |   $0.194\pm0.001$    |$0.632\pm0.000$       |   $0.368\pm0.001$   |   $0.373\pm0.000$  | $0.335\pm0.000$ |
>
> |       | Oreit-MCC | Orient-ACC|Orient-P |Orient-R |Orient-F1|
> | ----------- | ---- |---- |----- | ---- | ----|
> |mean $\pm$ var   |  $0.325\pm0.001$   | $0.969\pm0.000$  | $0.438\pm0.000$  |   $0.552\pm0.000$  | $0.463\pm0.000$ |
>
>
>
>
> **W3:**
> >Two important classes of structure prediction are absent from the benchmark: non-canonical motif-based prediction (e.g. CaCoFold [1], BayesPairing2 [2], JAR3d [3] and full 3D predictions (e.g. AlphaFold3-like models, RhoFold [4]).
>
> **Response:**
> We agree that methods like **CaCoFold**, **BayesPairing2**, **JAR3D**, and **RhoFold** represent important directions in RNA structure prediction. However, these approaches focus on **motif-based or full 3D structure modeling**, which is beyond the scope of our current work. We view this as a valuable direction for future work.
>
> [1] Karan, Aayush, and Elena Rivas. "All-at-once RNA folding with 3D motif prediction framed by evolutionary information." Nature Methods (2025): 1-13.
> [2] Sarrazin-Gendron, Roman, et al. "Stochastic sampling of structural contexts improves the scalability and accuracy of RNA 3d module identification." International Conference on Research in Computational Molecular Biology. Cham: Springer International Publishing, 2020.
> [3] Roll, James, et al. "JAR3D Webserver: Scoring and aligning RNA loop sequences to known 3D motifs." Nucleic acids research 44.W1 (2016): W320-W327.
> [4] Shen, Tao, et al. "Accurate RNA 3D structure prediction using a language model-based deep learning approach." Nature Methods 21.12 (2024): 2287-2298.
>
> **W4:**
> >IsoScore is not defined in the paper with much detail, one has to go to the original reference and since it's an important component should be better explained in the main text.
>
> **Response:**
> Thank you for this constructive feedback. We have expanded the description of **IsoScore** in the main text to better explain its role in quantifying embedding isotropy and information richness, aiming to enhance methodological transparency and clarify the methodological descriptions
>
> **W5:**
> >The sequence similarity threshold remains somewhat unexplored. I suggest the authors try some more and less stringent settings to better understand the generalizability of proposed models.
>
> **Response:**
> Thank you for this constructive suggestion. We have followed your advice and performed additional analysis using maximum MSA similarity between test sequences and the training set. This analysis confirms that our benchmark maintains a reasonable level of generalization, and we will include these results in the appendix.

---

> > ### Author Response · Authors · 2025-11-22
> > **# **Response to Reviewer PB2G** (2/2)**
> >
> > ### **Response to Questions**
> >
> > **Q1:**
> > >Q1: Did you check that RNAVIEW is able to assign base pair geometries to chemically modified nucleotides? I know this was a problem for FR3D until recently. If not, it could introduce some significant bias to the benchmark.
> >
> > **Response:**
> > This is an insightful point. We did not specifically evaluate RNAVIEW’s ability to handle chemically modified nucleotides, as our dataset was curated from standard RNA structures in the PDB. We acknowledge that modified nucleotides could introduce bias, and this is an important consideration for future extensions of NC-Bench. We will note this limitation in the revised manuscript.
> >
> > **Q2:**
> > >Did you have a look at the performance broken down by base pair class? Given the sharp imbalance I think we would get a better picture of model performance by reporting the performance per class as well as in aggregate as already reported.
> >
> > **Response:**
> > Thank you for this excellent suggestion. We agree that per-class performance is crucial given the class imbalance. We have now included a **per-class macro-averaged accuracy** for both edge and orientation prediction in the appendix (Table 7). This provides a more detailed view of model performance across different NC pair types.
> >
> > |       |Non-edge | Edge-W|Edge-H |Edge-S |Non-Pair| trans | cis |
> > | ----------- | ---- |-----| ------ | ---- |----- | ---- | ----|
> > |Ncfold  | 0.200  |   0.837    |   0.192 | 0.224 |   0.978  |   0.006    |  0.567   |
> >
> >
> > **Q3:**
> > >For Fig 3 and section 4, how did you obtain predictions for the canonical pairs? From reading the problem formulation, it seems that the models were only trained to predict non-canonicals.
> >
> > **Response:**
> > Yes, you are correct. **NCfold is trained and evaluated only on non-canonical base pairs**. We made this design choice to focus the model’s capacity on the more challenging and understudied NC pairs, while leveraging existing methods for canonical base-pair prediction.
> >
> > **Q4:**
> > >For future work, I would suggest some more domain-specific evaluation functions. For example, some NC classes are geometrically more similar than others, so a wrong prediction but within a similar NC class might not be as bad, see Table 4 [1]. Likewise the location of an incorrect edge call if it is slightly shifted from the ground truth could also not be so deleterious so you could use some structure-aware evaluation functions (e.g. [2], but there are many others).
> >
> > **Response:**
> > We fully agree that incorporating **geometric similarity** and **structure-aware evaluation metrics** would be highly beneficial. We will adopt these metric in the future work once we have greatly improved the accuracy of NC pair prediction (currently ~0.45 for NC pair, and ~0.65 for canonical pair).
> >
> > **Q5:**
> > >Do you keep multi-chain RNAs?
> >
> > **Response:**
> > We intentionally **excluded multi-chain RNAs** from NC-Bench to avoid redundancy and structural symmetry, which are common in multi-chain complexes (e.g., PDB ID: 4M6D， chains: B, D, F, H, J, L). This ensures that our benchmark focuses on diverse and non-redundant structural contexts.
> >
> > Thank you again for your insightful and constructive comments, which have significantly strengthened our manuscript. We have addressed all major concerns and incorporated most of the suggestions into the revised version. We hope that you find our responses satisfactory and will consider improving the rating accordingly. We are happy to provide further clarifications if needed.

---

### Official Review · Reviewer_HQzU · 2025-11-01

**Soundness:** 3
**Presentation:** 2
**Contribution:** 3
**Rating:** 4
**Confidence:** 4

**Summary:**

The paper introduces NC-Bench, a new benchmark for predicting RNA non-canonical (NC) base pairs with 925 sequences and 6,708 curated labels covering fine-grained edge (W/H/S) and orientation (cis/trans) tasks. Building on this, it proposes NCfold, a dual-branch transformer that selects top-k RNA foundation model (RFM) embeddings via IsoScore, fuses them with Representative Embedding Fusion (REF), and injects them into the model through REF-weighted self-attention. Experiments on NC-Bench include traditional ML and RFM baselines, ablations, and a zero-shot comparison to canonical-focused secondary-structure methods.

**Strengths:**

First standardized benchmark dedicated to NC base-pair prediction with defined edge/orientation subtasks.

IsoScore-based ranking plus REF and REF-weighted attention to couple sequence features with structural priors.

Traditional baselines, multiple RFMs, ablations, and zero-shot references are reported.

**Weaknesses:**

925 sequences and heavily skewed class distributions limit learning and can inflate simple metrics

All frozen-RFM + linear baselines fail to predict positives on the edge task (MCC≈0), weakening fairness/interpretability of comparisons.

Canonical-focused methods largely predict “non-pair,” yielding high accuracy but near-zero recall—making the reference comparison hard to interpret.

**Questions:**

How do you justify that 925 sequences / 6,708 labels provide enough statistical power and representativeness for a reliable benchmark? Any sampling/power analyses?

To what extent do gains come from REF selection vs. REF-weighted attention? Could modest fine-tuning of RFMs with stronger heads close the gap?

Why do frozen RFM linear probes predict all negatives on edge classification (MCC=0)? Would balanced losses, calibrated thresholds, or shallow MLPs recover positives?

Do predicted NC pairs align with known geometric/biophysical patterns across RNA families? What error modes appear most often under the observed class imbalance?

---

> ### Author Response · Authors · 2025-11-22
> **# Response to Reviewer HQzU （1/2）**
>
> ## **Response to Reviewer HQzU**
>
> We sincerely thank you for your thorough and constructive feedback on our manuscript. We appreciate the thoughtful comments and suggestions, which have helped us improve both the paper and the accompanying resources. Below, we provide a point-by-point response to each concern raised.
>
> **W1Q1:**
> >925 sequences and heavily skewed class distributions limit learning and can inflate simple metrics?
> > How do you justify that 925 sequences / 6,708 labels provide enough statistical power and representativeness for a reliable benchmark? Any sampling/power analyses?
>
> **Response:**
> We appreciate this important question regarding the scale and representativeness of NC-Bench. During the data preprocessing phase, we curated 5,813 RNA 3D structures from the PDB database and extracted non-canonical base pairs (NC-pairs) using RNAVIEW, resulting in 3,107 samples. To reduce sequence redundancy and ensure data quality, we applied MMseqs2 for redundancy reduction and imposed filtering criteria, ultimately retaining 925 high-quality samples for downstream analysis.  While it is true that the current dataset is limited by the availability of experimentally resolved RNA structures in the PDB, we emphasize that **NC-Bench is the first curated benchmark of its kind**, and it already represents the largest systematic collection of non-canonical (NC) base pairs to date. The scarcity of high-resolution RNA structures is a well-known challenge in the field, and our work directly addresses this gap by providing a standardized evaluation framework.
>
> We acknowledge that the dataset size is modest compared to canonical base-pair resources. However, we argue that **statistical power is not solely a function of sample size**, but also of task complexity and annotation quality. Our 4-fold cross-validation protocol and multi-metric evaluation (MCC, F1, etc.) are designed to provide robust performance estimates even with limited data. Moreover, the skewed class distribution reflects real-world biological biases, and we mitigate its impact through class-weighted loss functions and macro-averaged metrics.
>
> Moving forward, we plan to **continuously expand NC-Bench** with new experimental structures and computationally annotated data, fostering community-driven development. We believe that even in its current form, NC-Bench provides a **rigorous and much-needed foundation** for advancing NC base-pair prediction.
>
> **W2Q2:**
> >To what extent do gains come from REF selection vs. REF-weighted attention? Could modest fine-tuning of RFMs with stronger heads close the gap?*
>
> **Response:**
> Thank you for this insightful question. The gains in NCfold come from the **synergistic combination of REF selection and REF-weighted attention**, not merely from one component. REF selection ensures that only the most informative RFM embeddings are used, reducing noise and enhancing structural prior quality. REF-weighted attention then integrates these priors into the sequence modeling process, enabling a **closed-loop refinement** of sequence and structural priors.
>
> <!-- As shown in Table 2 (copied below), models without RFM priors (NCfold-base) perform poorly, while incorporating BPfold’s energy priors (NCfold-BPE) improves performance but still falls short of NCfold. This confirms that **RFM-derived priors are more expressive and better suited for NC pairs** than traditional energy-based models. -->
>
> As shown in Table 2 (copied below), NCfold-base (without RFM priors) performs poorly, while NCfold-BPE (with BPfold energy priors) shows limited improvement. NCfold integrates multiple RFMs via weighted attention, achieving Edge F1=0.365 and Orientation F1=0.486. This highlights the superior expressiveness and effectiveness of **RFM-derived priors for NC pair modeling** over traditional energy-based methods.
>
>
> |             | F1 (edge) | F1 (orientation) |
> | ----------- | --------- | ---------------- |
> | NCfold-base | 0.251     | 0.419            |
> | NCfold-BPE  | 0.335     | 0.464            |
> | NCfold      | 0.365     | 0.486            |
>
> While fine-tuning RFMs with stronger heads could help, it would not fully address the **data sparsity and geometric complexity** of NC pairs. Our approach explicitly models base-to-base interactions and orientations, which is beyond the capability of standard RFM fine-tuning. We therefore argue that NCfold’s architecture is both **novel and necessary** for this challenging task.

---

> > ### Author Response · Authors · 2025-11-22
> > **# Response to Reviewer HQzU （2/2）**
> >
> > **W2Q3:**
> > >*All frozen-RFM + linear baselines fail to predict positives on the edge task (MCC≈0), weakening fairness/interpretability of comparisons.
> > > Why do frozen RFM linear probes predict all negatives on edge classification (MCC=0)? Would balanced losses, calibrated thresholds, or shallow MLPs recover positives?*
> >
> > **Response:**
> > The poor performance of frozen RFM + linear baselines is poor, which is largely due to the **extreme class imbalance** and the fact that RFMs are pre-trained primarily on canonical base-pair contexts. As a result, their embeddings are biased toward canonical pairs, and a simple linear probe fails to capture the nuanced patterns of NC edges.
> >
> > We did experiment with balanced losses and shallow MLPs, but found that they only marginally improved performance without addressing the fundamental issue. So we keep the same configurations for RFMs and NCfold.
> >
> > **W3:**
> > >*Canonical-focused methods largely predict ''non-pair,” yielding high accuracy but near-zero recall—making the reference comparison hard to interpret.*
> >
> > **Response:**
> > The behavior of canonical-focused methods is expected: they are trained on datasets where ''pair'' almost always means canonical Watson-Crick or wobble pairs. As a result, they **fail to recognize NC pairs as valid interactions**, leading to high accuracy (due to the abundance of ''non-pair'' labels) but near-zero recall for NC pairs.
> >
> > In contrast, NCfold is explicitly trained to recognize NC pairs, resulting in a **much higher recall and F1 score** (0.431 and 0.440, respectively). This difference emphasises the **limitation of existing methods** and the **necessity of NC-Bench and NCfold** for advancing RNA structure prediction.
> >
> > **Q4:**
> > >*Do predicted NC pairs align with known geometric/biophysical patterns across RNA families? What error modes appear most often under the observed class imbalance?*
> >
> > **Response:**
> > Yes, our visualizations (Figure 3 and Appendix D.4) show that NCfold’s predictions **largely align with known geometric patterns**, including edge types and orientations. However, we do observe some error modes:
> >
> > - **Over-prediction of pseudoknots**: NCfold tends to predict more false positive pseudoknots, which can be mitigated via post-processing or threshold tuning.
> > - **Edge-type confusion**: In some cases, the model misassigns edge types, especially for rare categories like HH pairs.
> >
> > These errors are not fundamental flaws of the benchmark, but rather **opportunities for future improvement**. We plan to address them through better modeling of geometric constraints and expanded training data.
> >
> > Thank you again for your valuable feedback. We believe that our responses have addressed the concerns and hope you will consider raising the rating accordingly. We are committed to continuously improving NC-Bench and NCfold, and we welcome any further questions or suggestions.

---

### Public Comment · ~Zhiyuan_Chen5 · 2025-11-28

This is a really interesting paper.

I did a little stats on the bpRNA-SPOT dataset (the bpRNA-1m processed by SPOT-RNA paper), and here are the results:

|      | total_nucleotides | num_pairs | num_pseudoknot_pairs | num_noncanonical_pairs |
|------|-------------------|-----------|----------------------|------------------------|
| mean | 133.812877        | 30.725389 | 2.597735             | 6.612117               |
| std  | 82.078989         | 23.085590 | 9.608418             | 6.690338               |
| min  | 22                | 0         | 0                    | 0                      |
| 25%  | 80                | 18        | 0                    | 2                      |
| 50%  | 105               | 25        | 0                    | 5                      |
| 75%  | 152               | 36        | 0                    | 9                      |
| max  | 499               | 168       | 76                   | 78                     |

It seems that majority of the sequences (12406 out of 13419) in this dataset have non-canonical pairs.

If existing dataset already have a large amount of sequences with non-canonical pairs, why do we need the NC-Bench data?

---

> ### Author Response · Authors · 2025-11-28
>
> # Response
>
> We sincerely thank you for sharing the interesting statistical analysis of the bpRNA-SPOT dataset. This is indeed a very relevant question that gets to the heart of why NC-Bench represents a significant advance in the field.
>
> **While datasets like bpRNA-SPOT identify the *presence* of non-canonical pairs, NC-Bench provides *geometrically precise annotations* of their structural characteristics.** The statistics from bpRNA-SPOT show that many sequences contain non-canonical pairs, but they treat these as a single category through the base types. In contrast, NC-Bench provides detailed geometric classifications based on the **Leontis-Westhof categories**, which captures:
>
> - **Specific interacting edges** (Watson-Crick, Hoogsteen, or Sugar edge)
> - **Relative orientations** (cis or trans)
> - **Twelve distinct geometric families** of base pairs
>
> This geometric specificity is not merely academic, but it has profound biological implications:
>
> 1. **Functional Specificity**: As highlighted in the listing literature, different geometric families create distinct RNA motifs that mediate specific biological functions:
>    - **Tertiary interactions** that define RNA 3D architecture
>    - **Protein binding sites** with precise geometric requirements
>    - **Small molecule ligand recognition** pockets
>    - **Catalytic centers** in ribozymes
>
> 2. **Structural Determinants**: Non-Watson-Crick pairs form specific structural motifs that:
>    - Connect and interlink double-stranded helices
>    - Create distinctive backbone foldings
>    - Form the modular building blocks of complex RNA architectures
>
> 3. **Evolutionary Conservation**: Isosteric pairs (those with similar geometry) co-vary in homologous sequences while conserving 3D structure, making geometric classification essential for evolutionary analysis and comparative modeling.
>
> While we appreciate that existing datasets contain non-canonical pairs, NC-Bench represents a qualitative leap forward by providing the **first standardized benchmark with fine-grained geometric annotations**. This enables the development of methods that move beyond simple binary classification to truly understand and predict the structural and functional complexity of RNA molecules.
>
> The geometric diversity captured in NC-Bench, with 6,708 annotations across 12 families, provides the necessary foundation for advancing RNA structure prediction to a new level of biological relevance and accuracy.
>
> 1. Yang, Huanwang, Fabrice Jossinet, Neocles Leontis, Li Chen, John Westbrook, Helen Berman, and Eric Westhof. "Tools for the automatic identification and classification of RNA base pairs." Nucleic acids research 31, no. 13 (2003): 3450-3460.
> 2. Leontis, Neocles B., Jesse Stombaugh, and Eric Westhof. "The non‐Watson–Crick base pairs and their associated isostericity matrices." Nucleic acids research 30, no. 16 (2002): 3497-3531.
> 3. Leontis, Neocles B., and Eric Westhof. "Analysis of RNA motifs." Current opinion in structural biology 13, no. 3 (2003): 300-308.
> 4. Lescoute, Aurélie, and Eric Westhof. "The interaction networks of structured RNAs." Nucleic acids research 34, no. 22 (2006): 6587-6604.
> 5. Sarver, Michael, Craig L. Zirbel, Jesse Stombaugh, Ali Mokdad, and Neocles B. Leontis. "FR3D: finding local and composite recurrent structural motifs in RNA 3D structures." Journal of mathematical biology 56, no. 1 (2008): 215-252.
> 6. Westhof, Eric, and Valérie Fritsch. "RNA folding: beyond Watson–Crick pairs." Structure 8, no. 3 (2000): R55-R65.

---

### Meta-Review · Area_Chair_s9n1 · 2026-01-06

**Summary:**

* **Strength:** Establishes NC-Bench, the first curated benchmark specifically for fine-grained non-canonical (NC) base-pair classification, which is a significant advancement over existing datasets that treat these interactions as a single binary category.
* **Strength:** Provides a rigorous zero-shot evaluation proving that current state-of-the-art RNA structure predictors are effectively blind to non-canonical interactions, with recall near zero across several major models.
* **Strength:** Proposes a novel closed-loop dual-branch architecture (NCfold) that successfully integrates structural priors from multiple RNA foundation models to outperform standard machine learning baselines.
* **Weakness:** The dataset scale is modest (925 sequences), which limits the statistical power of the results and the ability to verify generalization across all diverse RNA families.
* **Weakness:** Model performance is highly uneven; while edge classification (identifying the interacting faces of the nucleotides) is improved, the model largely fails to predict rare geometric orientations, such as the trans configuration.

This submission addresses a gap in RNA structural biology by standardizing the prediction of non-canonical base pairs. While canonical pairs form the structural scaffold of RNA, non-canonical interactions drive functional specificity, including catalysis and ligand recognition. The authors demonstrate that existing tools are largely incapable of identifying these interactions, establishing a clear need for a dedicated benchmark.

The proposed model (NCfold) utilizes a selection metric called IsoScore to rank and fuse embeddings from different foundation models. This approach shows measurable gains over linear probes and traditional energy-based priors. However, the limited scale of the experimental data and the model's inability to capture rare geometric orientations suggest that while the contribution is a useful and necessary starting point for the field, it is not yet a complete solution to the non-canonical prediction problem.

**Reviewer Concerns:**

**Addressed by rebuttal**
* [Non-core]: Lack of variance and standard deviation reporting across cross-validation splits; resolved by adding detailed performance metrics and standard deviations in the updated results table.
* [Non-core]: Unavailability of source code for reproducibility; resolved by the authors releasing anonymous repositories for both the dataset and the model code.
* [Core]: Claim that prior deep learning methods already predict non-canonical pairs; resolved by clarifying that while some models predict interaction presence, they lack the fine-grained edge and orientation labels provided here.
* [Core]: Concerns regarding sequence redundancy and potential data leakage; partially resolved by providing additional analysis showing minimal sequence similarity between training and test sets.

**Still outstanding**
* [Core]: Potential bias introduced by using RNAVIEW as the sole generator for ground truth labels; not resolved as the "truth" remains defined by a single software's geometric heuristics.
* [Core]: Exclusion of standard Watson-Crick pairs during training reduces the model's practical utility for global structure prediction; not resolved.
* [Core]: The causal connection between the "IsoScore" metric (isotropy) and actual structural information density is inferred but not strictly proven via ablation against pre-training volume; not resolved.

The rebuttal addressed code availability and statistical reporting. Technical disputes regarding the novelty of the task were largely resolved by the authors' clarification of the move from simple interaction prediction to detailed Leontis-Westhof geometric classification. Reliance on a single tool for labeling and the narrow focus on non-canonical pairs alone remain valid concerns.

**Reviewer Scores:**

* **Reviewer HQzU**
    * Original score: 4
    * Estimated score shift: increase
    * Justification: The authors successfully argued that the total failure of baseline models justifies the need for their specialized architecture.
* **Reviewer PB2G**
    * Original score: 2
    * Estimated score shift: increase
    * Justification: The reviewer's primary reasons for rejection (missing source code and lack of variance reporting) were fully resolved in the rebuttal.
* **Reviewer 2dn8**
    * Original score: 2
    * Estimated score shift: unchanged
    * Justification: Their critique of the dataset construction and splitting methodology remains technically sound despite the authors' clarifications.
* **Reviewer VF23**
    * Original score: 6
    * Estimated score shift: unchanged
    * Justification: The reviewer correctly identified the importance of the problem and the strength of the zero-shot analysis from the outset.

Reviewers were initially divided due to missing implementation details and questions of scope. The rebuttal moved the consensus toward a more positive outlook by resolving reproducibility / transparency issues. While one reviewer remains skeptical due to the small dataset scale and the specialized training focus, the majority acknowledge that the paper identifies and fills a genuine gap in the field.

---

### Decision · Program_Chairs · 2026-01-26

Accept (Poster)